# Origin and role of the cerebrospinal fluid bidirectional flow in the central canal

Olivier Thouvenin[1,2†], Ludovic Keiser[3†], Yasmine Cantaut-Belarif[1], Martin Carbo-Tano[1], Frederik Verweij[4], Nathalie Jurisch-Yaksi[5,6], Pierre-Luc Bardet[1], Guillaume van Niel[4], Francois Gallaire[3]*, Claire Wyart[1]*

[1]Institut du Cerveau et de la Moelle épinière (ICM), Sorbonne Universités, UPMC Univ Paris 06, Inserm, CNRS, AP-HP, Hôpital Pitié-Salpêtrière, Paris, France; [2]ESPCI Paris, PSL University, CNRS, Institut Langevin, Paris, France; [3]Laboratory of Fluid Mechanics and Instabilities, École Polytechnique Fédérale de Lausanne, Lausanne, Switzerland; [4]Institute of Psychiatry and Neuroscience of Paris, Hôpital Saint-Anne, Université Descartes, INSERM U1266, Paris, France; [5]Kavli Institute for Systems Neuroscience, Centre for Neural Computation, The Faculty of Medicine, Norwegian University of Science and Technology, Trondheim, Norway; [6]Department of Clinical and Molecular Medicine, The Faculty of Medicine, Norwegian University of Science and Technology, Trondheim, Norway

**Abstract** Circulation of the cerebrospinal fluid (CSF) contributes to body axis formation and brain development. Here, we investigated the unexplained origins of the CSF flow bidirectionality in the central canal of the spinal cord of 30 hpf zebrafish embryos and its impact on development. Experiments combined with modeling and simulations demonstrate that the CSF flow is generated locally by caudally-polarized motile cilia along the ventral wall of the central canal. The closed geometry of the canal imposes the average flow rate to be null, explaining the reported bidirectionality. We also demonstrate that at this early stage, motile cilia ensure the proper formation of the central canal. Furthermore, we demonstrate that the bidirectional flow accelerates the transport of particles in the CSF via a coupled convective-diffusive transport process. Our study demonstrates that cilia activity combined with muscle contractions sustain the long-range transport of extracellular lipidic particles, enabling embryonic growth.

*For correspondence:
francois.gallaire@epfl.ch (FG);
claire.wyart@icm-institute.org
(CW)

[†]These authors contributed equally to this work

## Introduction

Precise control of flow in biological systems allows efficient transportation of critical molecules throughout an organism. In order to respond to cellular demands, passive diffusion in tissues can be sufficient as long as cells are close enough to a supply source. For organisms larger than a few hundreds of microns, diffusion alone is too slow to achieve supply of nutrients or signaling molecules to all cells in the body. Therefore, organisms have developed a large variety of flows to accelerate and orient the transport of key molecules. Blood flow is the main carrier of glucose and oxygen in most organisms and is mostly driven by the pressure gradient caused by the heart pump (*Fung, 2013*). Intestine, esophageal, and lymph flows rely on peristalsis, where muscles contractions drive the fluid in the direction of propagation of the contraction wave (*Fung and Yih, 1968*; *Goyal and Paterson, 2011*; *Hennig et al., 1999*; *Kim et al., 2017*; *Shapiro et al., 1969*). Other flows involve asymmetric beating of motile cilia or flagella, which are efficient motors that generate directional flows at small scales and mix fluids in biological systems (*Ferreira et al., 2017*; *McGrath et al., 2003*; *Shields et al., 2010*; *Supatto and Vermot, 2011*). Cilia-driven flows are critical for left/right asymmetry in the developing embryos of many species including humans (*Baker and Beales, 2009*), in

the Kupffer's vesicle in zebrafish (*Ferreira et al., 2017*; *Supatto et al., 2008*), and in the embryonic node in mammals (*Essner, 2005*; *Nonaka et al., 2005*; *Okada et al., 2005*). Cilia-driven flows can also be observed in the pronephros (*Kramer-Zucker et al., 2005*; *Obara et al., 2006*), respiratory tract (*Oldenburg et al., 2012*), and nasal cavity (*Reiten et al., 2017*). The motility of cilia in brain ventricles is also correlated with the cerebrospinal fluid (CSF) flow in zebrafish (*Grimes et al., 2016*; *Kramer-Zucker et al., 2005*; *Olstad et al., 2018*), African clawed frog (*Hagenlocher et al., 2013*; *Miskevich, 2010*) and rodents (*Faubel et al., 2016*).

CSF circulation plays various roles in the development of the brain and spinal cord. The CSF provides the hydromechanical protection of the brain (*Sakka et al., 2011*), transports nutrients, waste (*Sakka et al., 2011*; *Xie et al., 2013*), exosomes (*Bachy et al., 2008*; *Schneider and Simons, 2013*; *Sternberg et al., 2018*; *Tietje et al., 2014*; *Verweij et al., 2019*), and signaling molecules that impact neurogenesis and neuronal migration (*Lehtinen et al., 2011*; *Paul et al., 2017*; *Sawamoto et al., 2006*; *Silva-Vargas et al., 2016*). CSF circulation was recently shown to be critical for body axis formation in the embryo and spine morphogenesis in zebrafish juveniles and adults (*Grimes et al., 2016*; *Van Gennip et al., 2018*; *Zhang et al., 2018*). During embryogenesis, mutations that affect cilia polarity or motility and ciliogenesis lead to defects in transport along the rostro-caudal axis and body axis formation (*Boswell and Ciruna, 2017*; *Zhang et al., 2018*).

Historically, it has been generally assumed that CSF flow along the walls of all cavities must be driven by motile cilia. Nonetheless, it has been formally demonstrated only in the brain ventricles, where the direction of beating cilia indeed correlates with the direction of CSF flow (*Faubel et al., 2016*; *Olstad et al., 2018*). However, in the long cylindrical central canal, the dynamics of CSF flow is strikingly different than in the brain ventricles. We recently showed in 24–30 hr post fertilization (hpf) zebrafish embryos that the CSF flows bidirectionally (*Cantaut-Belarif et al., 2018*; *Sternberg et al., 2018*). Interestingly, body-axis defects observed in mutants with defective cilia appear by 30 hpf, that is, a few hours before cilia of the brain ventricles become motile and a directed CSF flow emerges in the ventricles (*Fame et al., 2016*; *Olstad et al., 2018*). This result suggests that proper dynamics of CSF flow in the central canal, rather than in the brain ventricles, is critical for embryonic development. Such a symmetrical bidirectional flow in a single channel is not common in biological systems.

To determine the mechanisms leading to bidirectional CSF flow in the central canal, we developed an automated method to quantify CSF flow in the central canal of zebrafish embryos. We first show that motile cilia are required to form the lumen of the central canal, as the canal has a reduced diameter in mutants with defective cilia. We combined experimental, numerical, and theoretical approaches to demonstrate the role of the spatially-asymmetric distribution of motile cilia in the generation of a bidirectional flow. We developed a general hydrodynamic model to account for the contribution of cilia, and show that this model can be applied as a general tool for many cilia-driven flows in confined environments. Additionally, we show experimental and theoretical evidence for an intricate relationship between local flow and long-range transport of particles in the CSF. We demonstrate that this bidirectional CSF flow in the central canal is enhanced by muscle contractions, which accelerate bidirectional transport of particles both rostrally and caudally. Finally, we investigate the global CSF circulation beyond the sole central canal. We show that the CSF flows unidirectionally from the diencephalic/mesencephalic ventricle to the entrance of the central canal, and that in vivo two-photon ablation of this connection leads to a reduction in size of embryos. It suggests a role for long-range CSF transport in growth during embryogenesis.

## Results

### Quantification of CSF flow in the central canal

In order to understand the mechanisms driving CSF flow in the central canal (CC), we first developed a thorough in vivo quantification of CSF flow in the CC. The obtained flow profiles will later serve as a basis to compare to our theoretical model. To establish quantitative and reproducible flow profiles along the dorsoventral (D-V) axis of the CC, we carried out an improved automated analysis inspired by our previous manual kymograph measurements (*Cantaut-Belarif et al., 2018*; *Sternberg et al., 2018*). By co-injecting TexasRed-Dextran (3,000 MW) with 20 nm-diameter green fluorescent beads in the diencephalic/mesencephalic ventricle (DV/MV), we measured both the volume of cavities and

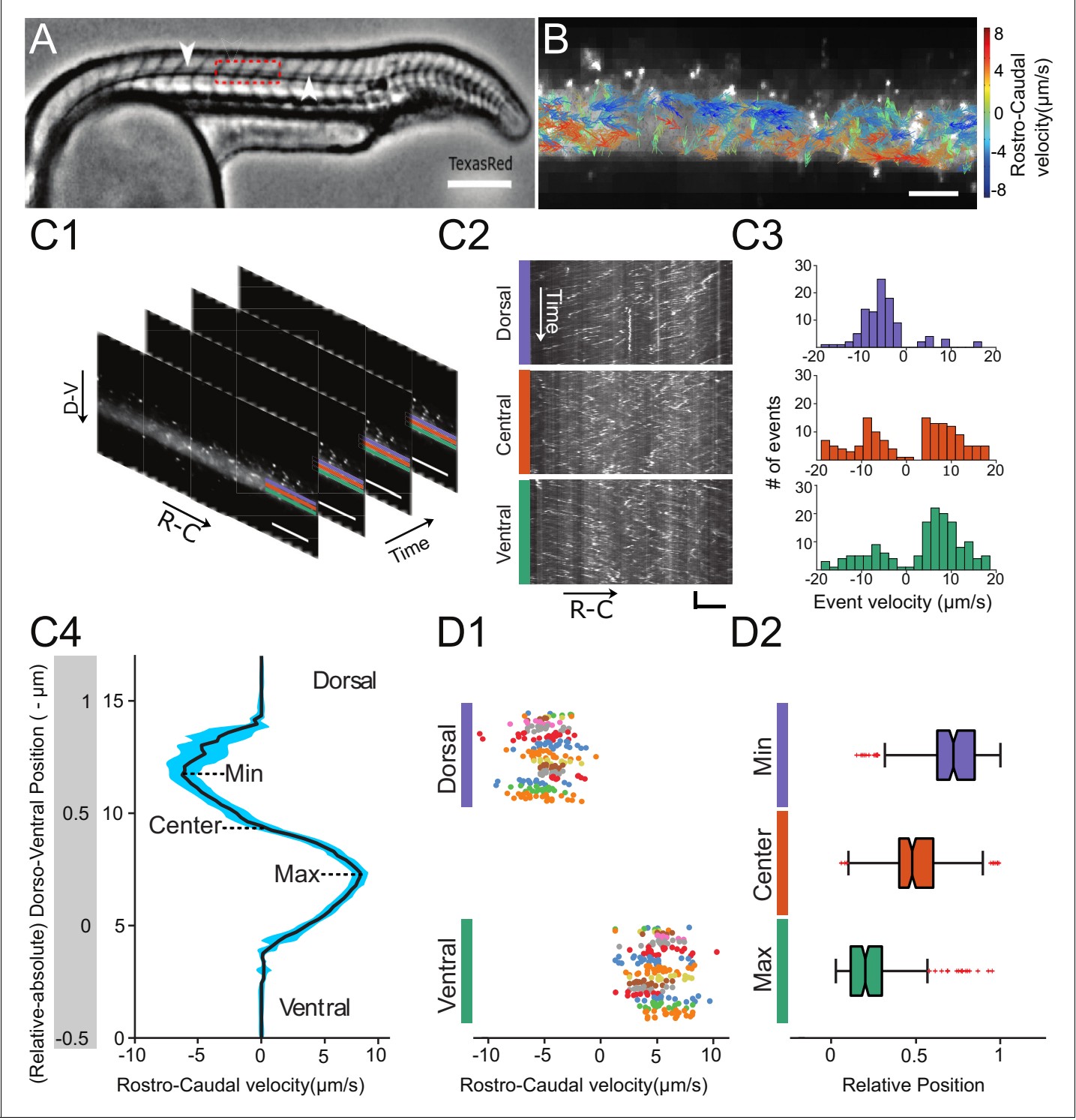

**Figure 1.** Automated CSF flow analysis in the central canal quantifies the bidirectional velocity profile in 30 hpf embryos. (**A**) Inverted contrast widefield image showing a 30 hpf zebrafish embryo injected with Texas Red: Texas Red fills all fluid-containing cavities, including ventricles, central canal (CC, white arrows), floor plate, somite boundaries, and blood vessels. (**B**) Instantaneous velocities of 20 nm fluorescent particles, as measured with a particle tracking velocimetry (PTV) algorithm, inside the central canal in a region corresponding to the red box in (**A**). The length, direction, and color of all arrows are coded according to the instantaneous bead velocity, ranging from −8 (blue) to 8 μm/s (red). (**C1**) Time series showing subsequent images of fluorescent beads injected in the DV and transported down the CC used to generate kymographs at different dorso-ventral (**D–V**) positions. (**C2**) Example kymographs computed in a dorsal, central, or ventral position in the CC. Each line represents the trajectory of one bead, projected along the rostro-caudal (**R–C**) axis. The slope of each trajectory gives a bead velocity projected onto the R-C axis. (**C3**) Histogram of velocities obtained at the

*Figure 1 continued on next page*

*Figure 1 continued*

three positions shown in (**C2**). (**C4**) Velocity profile (mean ± s.e.m. in blue) calculated for all D-V positions (expressed in absolute and relative position) in the CC for one WT embryo showing a maximum in the ventral side and a minimum in the dorsal side. (**D1**) Values of minimal (extremum in the dorsal part) and maximal velocities measured on 110 WT zebrafish embryos. Each dot represents the extremal values of one profile, and each color presents an experiment performed on siblings. 2 R-C positions are sampled per embryo. Extremal velocities are respectively 4.78 ± 0.79 µm.s-1 (ventral CC) and - 4.80 ± 0.82 µm.s-1 (dorsal CC). (**D2**) Values of relative D-V position of minimal, null, and maximal speed in the dorsal, center and ventral position in the CC (median positions are respectively 0.82, 0.53, and 0.29). Horizontal scale bar is 150 µm in (**A**), 15 µm in (**B, C1, C2**), and vertical scale bar is 5 s in (**C2**).

particle trajectories. The dextran rapidly diffuses throughout all fluid-filled cavities, including the CC (*Figure 1A*), enabling its visualization and assessment of the injection quality. Fluorescent nanobeads follow the bidirectional CSF flow (*Figure 1B*) but also undergo random Brownian motion due to their small size (*Video 1*). Bead trajectories also show the presence of a few regions of recirculation, commonly referred to as vortices (*Figure 1B*, *Video 1*). From the videos of fluorescent beads trajectories (*Video 1*, *Figure 1 C1*), we generated kymographs at various D-V positions in the CC (*Figure 1 C2, C3, C4*, and *Video 2*). The typical velocity profile is bimodal with positive flow (rostral to caudal) in the ventral side and negative flow (caudal to rostral) in the dorsal side, and reverses close to the center of the CC (relative D-V position = 0.53, *Figure 1D* from 110 wild type (WT) embryos each imaged in two rostro-caudal positions).

Automated kymographs show better reproducibility compared to particle tracking velocimetry (PTV) or particle imaging velocimetry (PIV), as discussed in Materials and methods section. Velocity profiles exhibit a maximal speed of 4.78 ± 0.79 µm.s$^{-1}$ (relative D-V position = 0.29 on average) ventrally and a minimal speed of - 4.80 ± 0.82 µm.s$^{-1}$ (relative D-V position = 0.82 on average) dorsally (*Figure 1B1 and B2*). The profiles are nearly symmetrical (relative D-V position at the zero-crossing = 0.53 on average). Similarly, the average speed throughout the CC is null (- 0.02 ± 0.15 µm.s$^{-1}$). This implies that the average CSF flow rate through a section of CC is null as well. According to Poiseuille's relationship, this refutes the hypothesis of a global pressure gradient as the unique driver of the flow in the CC.

## Central canal geometry and properties of the motile cilia

To build a theoretical model of CSF flow, essential parameters such as the canal geometry and the local cilia dynamics have to be quantified. The CC geometry, as estimated by TexasRed-Dextran (3,000 MW) injections, mostly corresponds to a cylinder (*Figure 2A1 and A2*) of diameter 8.9 ± 0.9 µm (129 WT embryos, *Figure 2A3*). The dynamics of motile cilia in the CC are investigated in 30 hpf *Tg(β-actin:Arl13b-GFP)* embryos in which GFP labels all cilia (*Borovina et al., 2010*; *Cantaut-Belarif et al.,*

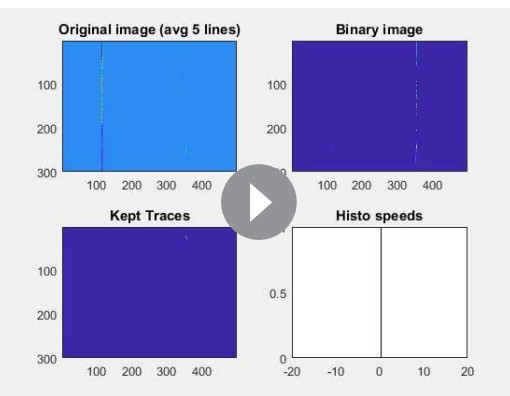

**Video 2.** Representative example of the automatic kymograph analysis of CSF flow in the central canal of a 30 hpf WT embryo. For all successive dorso-ventral positions, the unprocessed kymograph (top left panel) is binarized based on intensity threshold (top right panel). Only traces with adequate length, angle, and eccentricity are kept (bottom left panel) and their orientation was measured, converted in a corresponding bead velocity, and aggregated into a histogram (bottom right panel).
https://elifesciences.org/articles/47699#video2

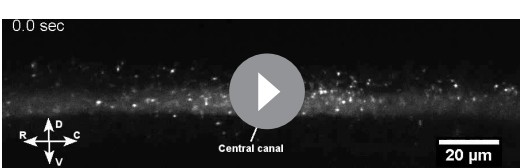

**Video 1.** Representative example of CSF flow observed in a 30 hpf WT embryo acquired with a spinning disk microscope at 10 Hz. The video is replayed at 35 frames per second (fps). Rostral (R) is to the left, ventral (V) is bottom.
https://elifesciences.org/articles/47699#video1

*2018*; *Sternberg et al., 2018*). Density and motility of cilia are higher in the ventral CC as previously reported (*Borovina et al., 2010*; *Cantaut-Belarif et al., 2018*; *Sternberg et al., 2018*). Movies recording the cilia beating illustrate the high density of cilia and diversity in ciliary length and beating frequency (*Figure 2B*, *Video 3* and *Figure 2—figure supplement 1*). While some cilia beat at high frequency (up to45 Hz, see Materials and methods) with a single frequency pattern, others beat at lower frequency (~13 Hz) with more complex and irregular patterns. An automated quantification of cilia beating frequency using temporal Fourier transform (*Figure 2B1*) revealed distinct spots of constant frequencies (*Figure 2B2*). These regions likely correspond to the area traveled by a single cilium during one beating cycle. For each spot, we extracted the main frequency, orientation, length and height (*Figures 2C1*, *3*, *4*). We separated dorsal and ventral motile cilia based on their relative position to the center of CC (see Materials and methods). Motile ventral cilia were four times more numerous than the dorsal ones. Ventral motile cilia were typically 5.8 μm long and beat at 38.1 Hz with a caudal tilt of 62.6° towards the tail (median of the absolute value, see Materials and methods). This cilia length corresponds to a height of 2.7 μm in the CC, representing a bit less than half of the CC on the ventral side. We observed a caudal tilt, consistently with previous reports (*Borovina et al., 2010*; *Cantaut-Belarif et al., 2018*). The beating frequencies we measured were higher than the 12–15 Hz previously reported in zebrafish CC (*Kramer-Zucker et al., 2005*) (see Materials and methods for a possible explanation), but are consistent with beating frequencies of ependymal cilia located in the brain ventricles of rats (*Smith et al., 2012*) and zebrafish larvae (*Olstad et al., 2018*; *Smith et al., 2012*). Dorsal motile cilia were sparse and evenly distributed along the rostro-caudal axis: they beat with a slightly smaller central frequency (~32 Hz, *Figure 2C2*), but exhibited similar properties to ventral cilia otherwise (see *Figure 2C3*, *4*, caudal polarization of 59.7 °, length of 5.5 μm).

## The asymmetric distribution of beating cilia generates a bidirectional flow

The CC is roughly shaped like a cylinder (*Figure 2A*) and contains motile cilia that line primarily the ventral wall (*Figure 2B*). The broad distribution of cilia beating frequency and cilia length, as well as the high density of cilia in the CC that could interact with each other and thereby introduce non linearities, would make it extremely challenging to model the effect of each individual cilium beating. As a first approximation, since the CSF bidirectional flow does not vary over time and over rostro-caudal axis (*Figure 1 C4*), we built a two-dimensional model that homogenizes cilia contribution as a constant force. The model divides the CC into two regions of equal thickness $h = d/2$ (*Figure 3 A1, A2*). It is assumed that the influence of dorsal cilia can be neglected, while active ventral cilia generate a constant volume force $f_v$ within the ventral region. Dimensional analysis gives $f_v = \alpha \mu f / h$, where $f$ is the cilia mean beating frequency (40 Hz), μ is the CSF dynamic viscosity and α is a numerical factor of order unity. The Stokes equation controls flow at small scales as inertial terms are negligible in comparison to viscous terms, and simplifies in dorsal and ventral regions as:

$$dP/dx = \mu \, \partial_y^2 v_{dorsal}, \tag{1}$$

$$dP/dx - f_v = \mu \, \partial_y^2 v_{ventral} \tag{2}$$

where $x$ and y represent the coordinates respectively in the rostro-caudal and dorso-ventral directions. $dP/dx$ is the common pressure gradient applying in both regions, and $\partial_y^2 v_{dorsal}$ is the second y-derivative of the rostro-caudal velocity $v_{dorsal}(y)$. The derived flow obeys two parabolic profiles:

$$v_{dorsal}(y) = \frac{dP/dx}{\mu}\big(y^2/2 - y_B y - d^2/2 + y_B d\big), \tag{3}$$

$$v_{ventral}(y) = \frac{dP/dx - f_v}{\mu}\big(y^2/2 - y_A y\big) \tag{4}$$

where $y_A$ and $y_B$ are two integration constants to be determined, and $dP/dx$ is the pressure gradient. Matching velocity and shear rate at the boundary between dorsal and ventral regions ($y = h$), two

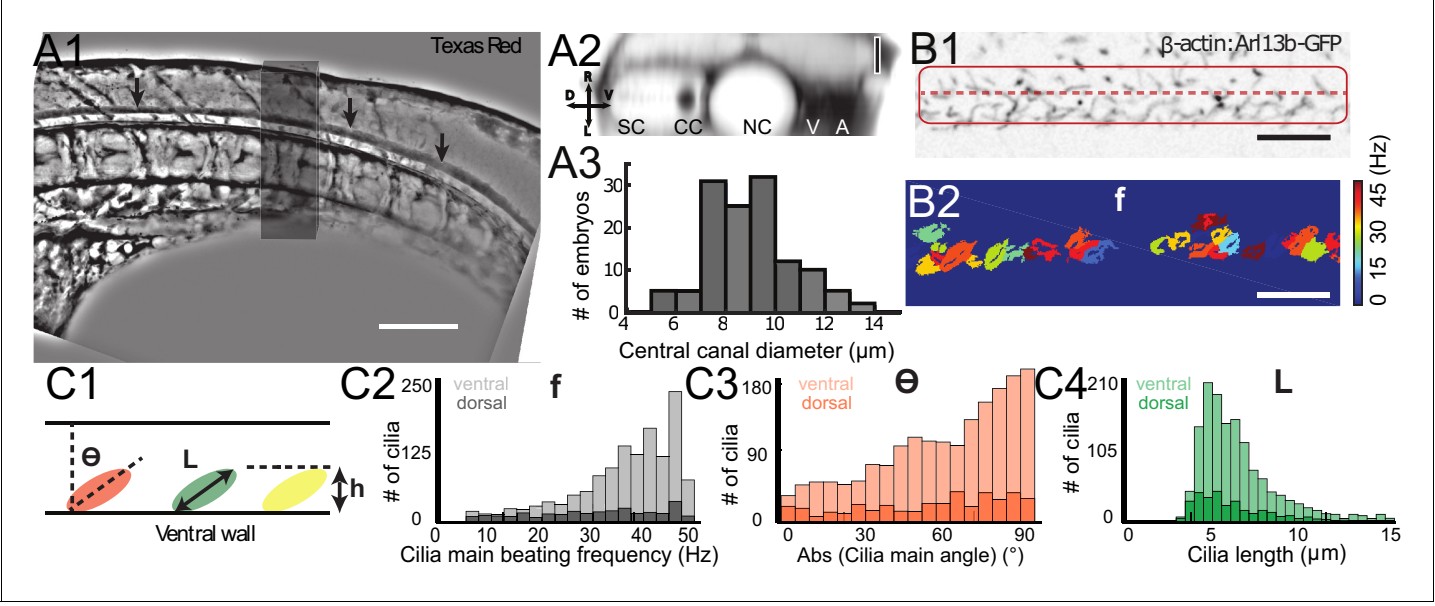

**Figure 2.** Geometry of the central canal and properties of motile cilia in 30 hpf embryos. (**A1, A2**) Central canal filled with TexasRed injected in the brain ventricles shown laterally from maximal Z-projection stack of 75 inverted contrast images acquired with 1 µm Z-step (**A1**) and corresponding vertical profile (**A2**, 5x rescale on the axial dimension enabled similar pixel size in X/Y and Z): SC, spinal cord, CC, central canal (black arrows in (**A1**)), NC, notochord, V, vein and A, artery. (**A3**) Histogram of central canal diameter from n = 129 WT embryos, with a median value of 8.9 ± 0.9 µm (mean ± 0.5 s.t.d.). (**B1, B2**) Lateral view in the CC where cilia are labeled with GFP in *Tg (β-actin:Arl13b-GFP)* transgenic embryos (**B1**) used to draw a map of local main beating frequency (**B2**) from the fluorescence signal obtained by calculating the local Fourier transform. Regions of constant frequency are color-coded and most likely correspond to single cilium beating. (**C1**) Scheme presenting some parameters of interest extracted from regions of constant frequency obtained in (**B2**) and fitted as ellipses to obtain their mean orientation Θ, their major axis (**L**), associated with the cilia maximal length, and their height (**h**), that is their projection along the D-V axis. The latter parameter gives the portion of central canal occupied by beating cilia. (**C2, C3, C4**) Histograms of parameters extracted in many regions of constant frequency of n = 89 *Tg(β-actin: Arl13b-GFP)* 30 hpf embryos, for a total of 1704 motile cilia (298 dorsal cilia and 1406 ventral cilia) analyzed in total. A central frequency of 38.1 Hz (median frequency with s.d 10.0 Hz) (**C2**), a mean absolute orientation |Θ| 62.6° (s.d. 23.4°) (**c3**), a length L of 5.8 µm (s.d. 2.0 µm) (**C4**) as well as a beating height of 2.7 µm (s.d. 1.9 µm) (not shown) were found for ventral cilia. Dorsal motile cilia have a central frequency of 32.3 Hz ± 12.3 Hz, a caudal orientation of absolute value 59.7°± 25.8° and a length of 5.5 µm ± 3.0 µm. Scale bar is 50 µm in (**A1**), and 15 µm in (**A2, B1, B2**).

The online version of this article includes the following figure supplement(s) for figure 2:

**Figure supplement 1.** Motile cilia beating pattern in the central canal of 30 hpf embryos.

equations can link these three unknowns. To solve this system of equations, a third relation is provided by the null-flux condition across the diameter of the central canal as observed in vivo:

$$\int_o^h v_{ventral}(y)dy = -\int_h^d v_{dorsal}(y)dy. \tag{5}$$

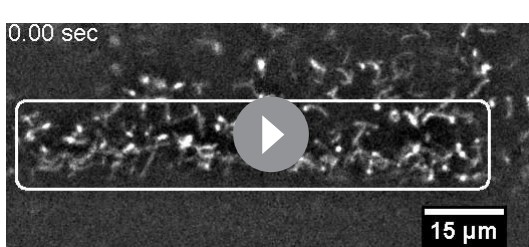

**Video 3.** 100 Hz video of the central canal where cilia are labeled with GFP in *Tg (β-actin: Arl13b-GFP)* transgenic embryos. The video is replayed at 10 fps.
https://elifesciences.org/articles/47699#video3

In the particular case of central canal geometry, the ventral ciliary region occupies about half of the diameter that is $h = d/2$. This enables to find: $y_A = d/4$, $y_B = 3d/4$ and $dP/dx = f_v/2$, and the velocity profile adopts this symmetrical biparabolic form:

$$v_{dorsal}(y) = \frac{dP/dx}{\mu}\left(y^2/2 - 3yd/4 + d^2/4\right) \tag{6}$$

$$v_{ventral}(y) = \frac{dP/dx - f_v}{\mu}\left(y^2/2 - yd/4\right) \tag{7}$$

More details concerning the derivation of the

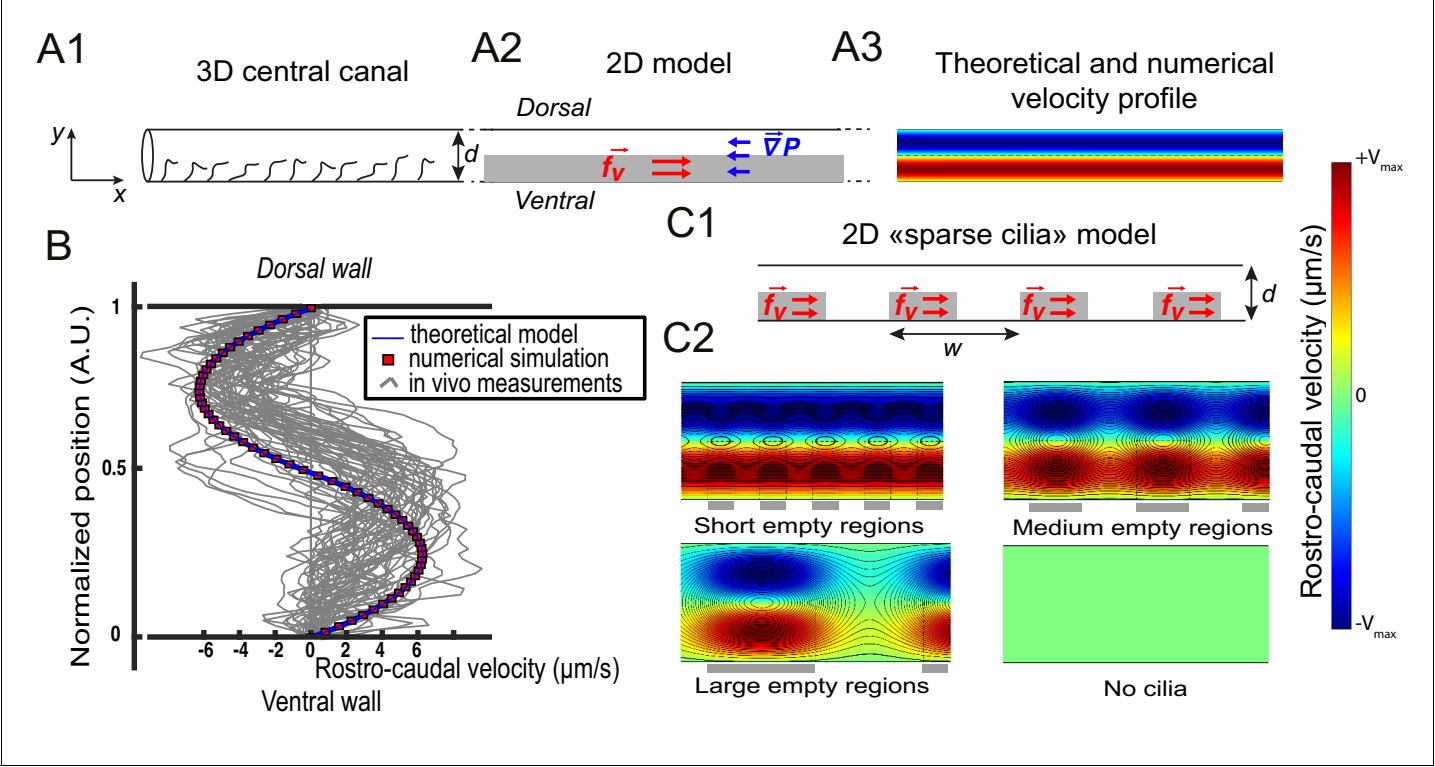

**Figure 3.** Theoretical and numerical results explain how motile cilia generate local CSF flow in 30 hpf embryos. (**A1**) Schematic of the three-dimensional central canal that can be reduced to a quasi-cylinder whose ventral wall is paved by regularly positioned motile cilia, and whose dorsal wall is composed of more loosely distributed passive cilia. (**A2**) The model reduces the geometry to a bidimensional channel of thickness d, composed of two main regions occupying the ventral and dorsal halves of the channel. In the ventral region, the cilia contribution is modeled as a constant bulk force. A constant gradient of pressure, present in the whole central canal, opposes the cilia-induced force and ensures a global zero flux on a cross section. (**A3**) Numerically and theoretically calculated velocity maps, where dark blue represents highest rostral velocities (negative values), while dark red holds for the highest caudal velocities (positive values). (**B**) Numerical (red squares) and theoretical (blue line) velocity profiles, composed of two parabolas matched at the center of the cross-section (Relative position = 0.5). They are convincingly compared to a superimposition of in vivo measurements obtained on 57 WT siblings. (**C1**) 'Sparse cilia' model, accounting for the spatially inhomogeneous defects in cilia structure or motility (as observed in some mutants with defective cilia). The model reduces these defects to regions in the ventral side that are fully passive and where no force applies. Numerical simulations enable to derive the flow with an arbitrary distribution of these passive regions. (**C2**) Maps of characteristic streamlines obtained for various aspect ratios w/d of the repeat units. Small passive regions show most of the streamlines parallel to the walls. A central recirculation, or vortex, is gradually more present in between cilia regions as the lateral extension of the passive region increases. The corresponding distribution of pressure in the central canal for these different cases is represented in *Figure 3—figure supplement 1*.

The online version of this article includes the following figure supplement(s) for figure 3:

**Figure supplement 1.** Additional information regarding our cilia-driven model and its consequences on local CSF flow and recirculation spots.

velocity profile can be found in the Appendix 1. Note that this symmetry in the velocity profile, represented in *Figure 3A3*, is only obtained for the particular case $h = d/2$.

With a unique fitting parameter α that controls the amplitude of the constant volumetric force chosen as 0.5, (leading to $f_v = 4000$ N/m³), we obtain a symmetric bimodal flow with extreme velocities equal to ±5 μm/s (*Figure 3B*), and an axial pressure gradient $dP/dx = 2000$ N/m³. We additionally solved the 2D Stokes equation by Finite Element Method numerical simulations (Comsol Multiphysics software). Theoretical and numerical predictions are in good agreement with the 57 velocity profiles measured in vivo and tested here (*Figure 3B*).

The model is further refined to account for the spatial heterogeneity in the cilia distribution in the ventral region. We designed spots of constant volume force $f_v$ regularly distributed along the ventral wall of CC with a period $w$ (*Figure 3C1*), and added some regions without cilia in the ventral side. With this 'sparse cilia' model, numerical simulations show that recirculation regions appear between these spots (*Figure 3C2*). The amplitude of the recirculation regions is more pronounced as $w$ increases. For large passive regions of rostro-caudal extension $w$ larger than the diameter of the CC

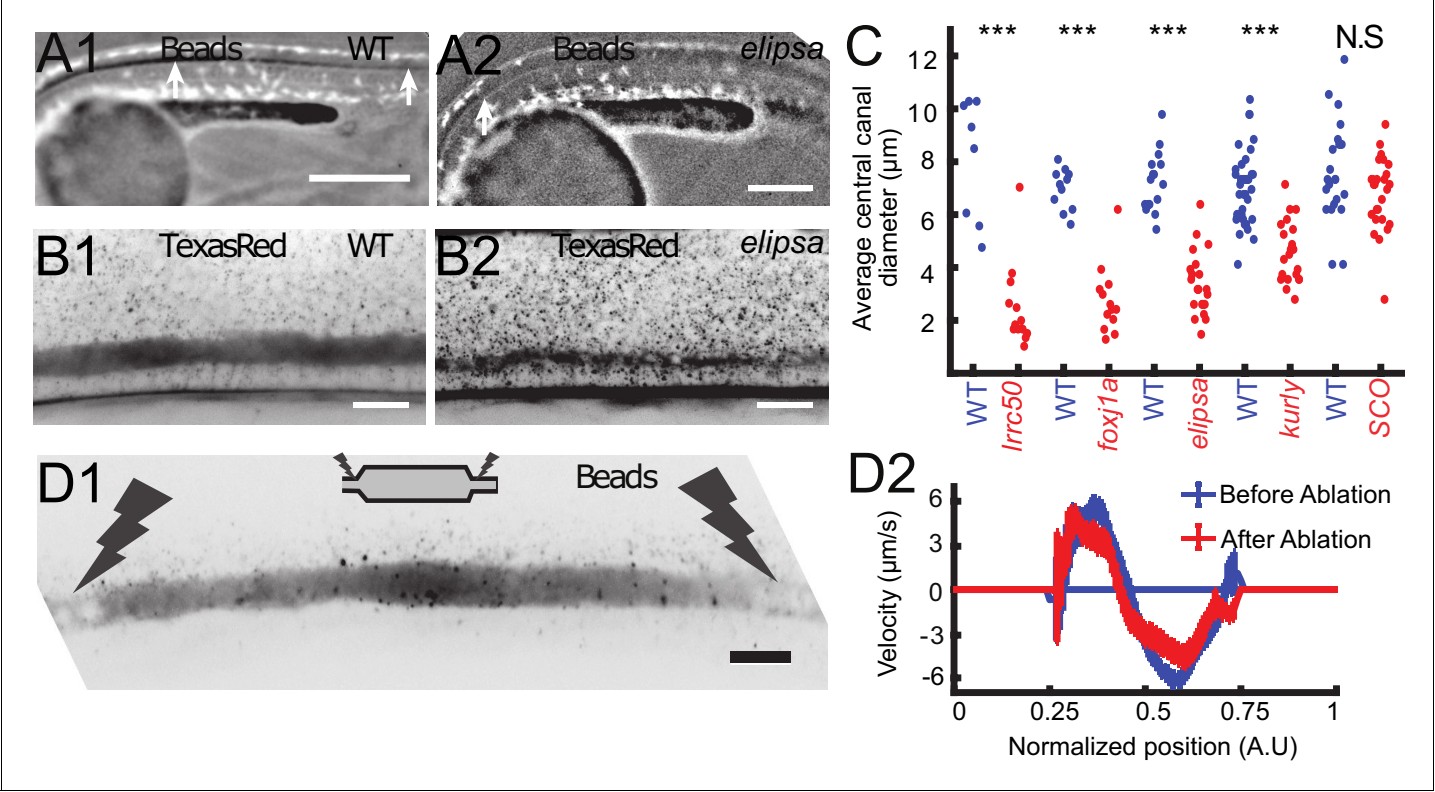

**Figure 4.** Cilia beating controls long-range transport, CC architecture, and local flow in zebrafish embryos. (**A1, A2**) Progression of 20 nm fluorescent beads inside the CC 2 hr post injection (two hpi) in the diencephalic/mesencephalic ventricle in a WT 30 hpf embryo (**A1**) and in a ciliary mutant (*elipsa*) sibling (**A2**). The progression front is demarcated by the rightmost white arrow in (**A1**) and (**A2**), showing that beads barely propagate in central canal in ciliary mutants. (**B1, B2**) Central canal geometry in a WT embryo (**B1**) and in a ciliary mutant (*elipsa*) sibling (**B2**). The central canal was filled with Texas Red, and is displayed as the maximal projection image from a Z-stack of 30 images around CC. The CC appears collapsed in an *elipsa* mutant. (**C**) Quantification of CC diameter for different ciliary mutants (*lrrc50, foxj1a, elipsa, kurly*) and one curled down mutant with intact CSF flow (*SCO*) (red dots, right side) versus their WT siblings (blue dots, left side). (**D1**) Time average image of 20 nm fluorescence beads in inverted contrast of a central canal region where photoablation was performed on both sides (dark lightning bolt), closing the canal. (**D2**) Quantification of local CSF flow in the CC region in (**D1**) before (blue) and after (red) photoablation. CC geometry remains intact in the central region (**D1**), as well as the flow profile (**D2**). Scale bars represent 200 μm in (**A1, A2**), and 15 μm in panels (**B1, B2, D1**). P-values in (**C**) are respectively $2.2 \times 10^{-6}$, $3.1 \times 10^{-9}$, $3.5 \times 10^{-6}$ $1.82 \times 10^{-8}$, 0.096 from left to right. The values for control WT siblings from all experiments are not statistically different.

The online version of this article includes the following figure supplement(s) for figure 4:

**Figure supplement 1.** Characterization of cilia in the central canal of *foxj1a*[nw2/nw2] mutant embryo.

(*w>d*), all the streamlines are located within the vortices, and no directed flow is observed between the active regions. This model thus predicts that vortices appear due to the alternation of active and passive regions. Note that the presence of passive regions also decreases the pressure in the CC proportionally to the fraction of passive regions (*Figure 3—figure supplement 1A*), a direct consequence of the linearity of the Stokes equation. We can further show that differences in beating frequency between active neighboring cilia lead to the emergence of the same recirculation regions (*Figure 3—figure supplement 1D*).

## Motile cilia control the long-range CSF transport and maintain the CC structure

To verify that asymmetric cilia beating is solely responsible for the bidirectional flow in vivo as proposed by our model, we aimed at measuring the CSF flow in several mutant embryos for which either ciliogenesis or ciliary motility is affected; The *traf3ip*[tp499] mutation, referred to as *elipsa* (*Omori et al., 2008*), affects a protein involved in the intraflagellar transport (IFT), which leads to early cilia degenerescence at ~30 hpf. The *dnaaf1*[tm317b] mutation, corresponding to the gene

formally called *lrrc50* (*van Rooijen et al., 2008*), leads to a complete loss of cilia motility by destabilizing the ciliary axoneme (*van Rooijen et al., 2008*). The *cfap298*$^{tm304}$ mutation, referred to as *kurly* (*Brand et al., 1996*), affects in a thermosensitive manner a cytoplasmic protein necessary for cilia motility and polarity (*Jaffe et al., 2016*). This protein is partly inactivated at the temperature at which our experiments were conducted (28°C, *Jaffe et al., 2016*). The *foxj1a*$^{nw2}$ newly-generated mutation (see Materials and methods and characterization in *Figure 4 – figure supplement 1*) reported here for the first time affects a transcription factor, which does not affect primary cilia but cilia whose tubulin is glutamylated (*Figure 4—figure supplement 1E1, E2*). Cilia with glutamylated-tubulin in the CC of WT embryos are caudally-polarized and ventrally-located and most likely correspond to motile cilia (*Figure 4—figure supplement 1E1, E2*).

In all these mutants with defective cilia, we injected beads in the DV/MV, but we could not observe the beads transported inside the CC. In anaesthetized mutant embryos, no particles entered the CC even several hours post injection, as illustrated here in *elipsa* embryos (*Figure 4A1 and A2*). This observation is consistent with previous reports (*Cantaut-Belarif et al., 2018*; *Zhang et al., 2018*) showing that long-range transport of beads down the central canal is altered in mutants with defective cilia. To account for this reduced transport, we verified whether the CC geometry was affected in embryos with defective cilia. Surprisingly, we observed that the CC diameter was dramatically reduced in all mutants tested: *elipsa, lrrc50, foxj1a*$^{nw2}$, and *kurly* (*Figure 4B,C*). Although the CC forms in these mutants, it is flattened along the D-V axis and not as straight as control siblings (*Figure 4B1, B2 and C*). In order to evaluate whether these features could be due to the convex body curvature of the embryos, we analyzed the CC structure of mutant embryos *scospondin*$^{icm15/icm15}$, which are also curled down but exhibit normal cilia activity (*Cantaut-Belarif et al., 2018*). For these mutants, we did not measure any difference in the CC diameter. This observation suggests that cilia integrity and motility are critical to properly form the CC and instruct its diameter and straightness.

As exogenous beads are not transported down the CC in anaesthetized mutants with defective cilia, we could not measure the local CSF flow, and hence test whether the cilia beating is directly responsible for CSF bidirectional flow. However, we noticed that allowing mutant embryos to spontaneously twitch (by not anesthetizing them before imaging) facilitated the transport of beads down the CC (see next section for the characterization of the effect of muscle contraction on CSF flow). By using the effect of muscle contractions, we were able to get beads down the CC in mutants with a moderate reduction of the CC diameter, such as *elipsa* and *kurly*. Fluorescent beads in the CC of *eli-*

*psa* and *kurly* mostly undergo random Brownian motion, but some allow to measure a residual flow with many recirculation zones (*Video 4*), in good agreement with the predictions of the sparse cilia model.

It is also predicted that the bidirectional CSF flow in the CC is generated locally and does not rely on a global pressure gradient generated in the ventricles. This is confirmed by multiple observations: i) even after large regions without cilia and local flow, the CSF flow in more caudal regions can show a bidirectional flow, as long as motile cilia beat locally in the ventral region (as predicted by *Figure 3C*). ii) When a portion of the CC of around 200 μm in length is isolated by using photoablation to block flow on both extremities (see Materials and methods and *Figure 4D1*), the canal kept its diameter at the center and the CSF still flowed bidirectionally (*Video 5*) with a similar velocity profile (*Figure 4D2*).

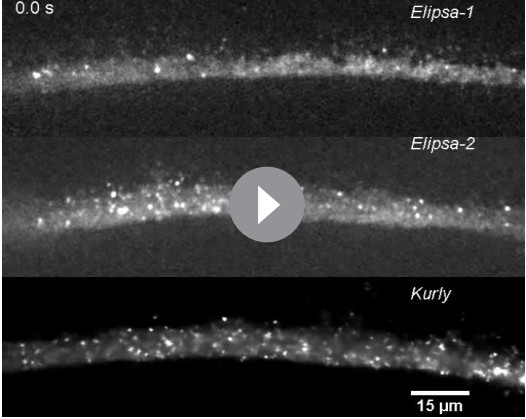

**Video 4.** Representative examples of CSF flow observed in intermediate ciliary mutants (*elipsa* mutants in the two top rows, and *kurly* mutant in the bottom row), showing either purely Brownian motion, or bidirectional flow with several vortices. The raw data were acquired at 10 Hz, and the video is replayed at 20 fps.
https://elifesciences.org/articles/47699#video4

## Muscle contraction transiently enhances CSF flow

So far, we mainly restricted our study to anaesthetized zebrafish embryos. However, muscle contractions can also generate several biological flows (*Blatter et al., 2016*). Recent results demonstrated that muscle contraction can create a massive transient flow in the brain ventricles of zebrafish embryos (*Olstad et al., 2018*). Consequently, we investigated CSF flow and transport on embryos without anesthesia, which have spontaneous contractions of the body wall muscles (spontaneous twitches). Animals were restrained in agar and contractions were identified from the notochord displacement (white arrow) on the transmitted image (*Figure 5A*). Following each contraction, rapid displacements of CSF (too fast to be properly characterized) were observed in either the caudal or rostral direction, followed by a slower counter-flow in the opposite direction, whose velocity decayed until the original bidirectional flow is recovered (*Figure 5B and C*, *Video 6*, *Video 7*). The average CSF flow throughout the entire CC section is correlated with the instantaneous muscle contractions (*Figure 5D*) and showed a sudden increase after contractions. By comparing the increase in CSF flow over 5 s after each contraction with the contraction strength, we observed a robust flow rate increase after each contraction (*Figure 5E*; 204 contractions from 22 WT embryos). However, we did not find any correlation between the flow direction and estimates of the contraction direction or strength. Body contractions also enhance transport of beads in mutants with defective cilia, as mentioned in the previous section (*Video 4*). As no significant difference in the CC diameter was detected during the muscle contractions, we propose that the reported enhancement of CSF flow is not due to a local variation of the CC diameter, but instead due to a change in the volume of the brain ventricles that occurs during muscle contraction at larval stage (*Olstad et al., 2018*).

## Bidirectional long-range transport of particles in the CC is accelerated by CSF flow

We characterized above the local CSF flow in the CC, but did not investigate its ability to transport particles over long distances, despite its importance for embryonic development (*Chang et al., 2016*). The rest of the study will focus on long-range circulation of CSF, first in the CC, and then in the entire CSF circulatory system. At the micrometric scale of the central canal, Brownian diffusion can be sufficient to spread molecules without requiring a flow advection. However, larger particles (of diameter larger than about 10 nm) are less sensitive to Brownian motion and the presence of a flow can dramatically increase their transport efficiency throughout the embryo body. It is at the crossover between these two qualitatively different transport phenomena that the influence of the bidirectional flow in the CC will be characterized.

Here, we first demonstrate experimentally that CSF long-range transport of nanoparticles has characteristics of both passive diffusion and active transport. The particle transport follows an apparent diffusive law, as the propagation front of nanoparticles progresses with the square root of time (*Zhang et al., 2018*) and this propagation is found to be slower for larger particle size (*Figure 6A1 and A2*). On the other hand, the propagation front of 20 nm-diameter fluorescent beads is accelerated by muscle contractions, when the local CSF flow velocity is increased (*Figure 6A2 and A3*). An estimation of orders of magnitude also suggests that CSF long-range transport originates from a combination of pure diffusion and flow advection. Indeed, 20 nm-diameter fluorescent beads injected in the DV/MV are found to travel through the entire length of CC (~2 mm) in about two hours (*Figure 6A2*), far from the timescales expected for passive diffusion (over one day) or pure ballistic transport at the speed of the flow (with V = 5 μm/s, this would lead to less than 10 min).

To account for these observations, we developed theoretical and numerical calculations according to CC geometry. We modeled the transport of particles by including an advection term in the diffusion equation. In a bi-dimensional canal (*Figure 3A2*), the transport equation can be written as:

$$\partial_t c \, + \, v(y)\partial_x c = D\left(\partial_y^2 c + \partial_x^2 c\right), \tag{8}$$

where $c$ is the local concentration of particles, and $\partial_t, \partial_x$ and $\partial_y$ are respectively the time and spatial derivatives in the rostro-caudal ($x$) and dorso-ventral ($y$) directions. $v(y)$ is the velocity profile (*Equations 6, 7*), and $D$ is the Brownian diffusion coefficient of particles in CSF. For a spherical particle of radius $r$, the Stokes-Einstein relation gives:

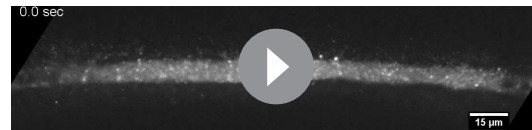

**Video 5.** 10 Hz video of 20 nm beads trapped between two ablated regions in central canal of a 30 hpf zebrafish embryo with spinning disk microscopy. The video is replayed at 35 fps.
https://elifesciences.org/articles/47699#video5

$$D = \frac{k_B T}{6 \pi r \mu} \qquad (9)$$

where $T$ is the temperature and $k_B$ is Boltzmann's constant. As described by *Taylor (1953)* and *Aris (1956)*, the presence of velocity gradients increases the effective diffusivity of solutes due to a coupling between classical Brownian diffusion and flow advection. Over long distances, transport still follows a diffusive process, but with an effective diffusivity enhanced by the shear flow (*Aris, 1956*; *Taylor, 1953*). From *Equation 8*, we performed the theoretical derivation of this effective diffusivity $D_{eff}$ for particles of arbitrary size in the presence of a bidirectional flow (in

**Figure 5.** Muscle contraction transiently increases local CSF flow in the central canal. (A) Three successive images showing the time derivative of the transmitted image of a zebrafish embryo during muscle contraction. A fast contraction in one direction is followed by a slower decay back to the initial position, during which the notochord comes back more slowly to its original position (white arrows). (B, C) Dorsal (B) and ventral (C) kymographs obtained after spinning disk imaging of injected 20 nm beads. Contractions can be identified as the sharp horizontal lines and are emphasized by the four black asterisks. (D) Average volumetric flow across one section of the central canal versus time (cyan) for one representative embryo. The embryo instantaneous displacement (magenta) is superimposed. Contractions are also emphasized by dark asterisks and a gray background. (E) Quantification of the integral of the average volumetric flow during 2.5 s after each of the 204 contractions from n = 22 WT embryos. These values are plotted versus the contraction strength, that is the integral of intensity derivative during the time of contraction. The dataset is separated into positive flows (red dots), in which flow after contraction is directed towards the tail, and into negative flows (blue dots), in which CSF flow is directed towards the head. Horizontal scale bar is 15 μm in (A, B, C). The vertical bar in (B, C) is 4 s.

**Video 6.** One representative example of CSF flow change after contraction in the central canal, acquired at 10 Hz with spinning disk microscopy. The video is replayed at 25 fps.
https://elifesciences.org/articles/47699#video6

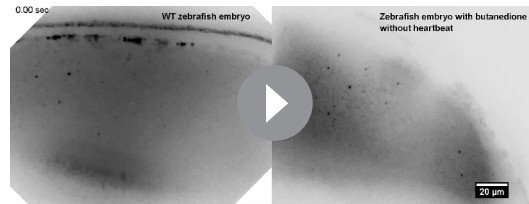

**Video 7.** A second representative example of CSF flow change after contraction in the central canal, acquired at 10 Hz with spinning disk microscopy. The video is replayed at 25 fps.
https://elifesciences.org/articles/47699#video7

Appendix 2, and *Appendix 2—figure 1*), based on the derivation done by Bruus for a unidirectional Poiseuille flow (*Bruus, 2008*). This leads to:

$$D_{eff} \approx D\left(1 + \frac{P\acute{e}^2}{24}\right) \tag{10}$$

where $\mathrm{PO} = Vd/D$ is the dimensionless Péclet number, comparing the characteristic times for spanwise diffusion $d^2/D$ and convection $\mathrm{d}/V$. The particle transport process in the CC thus follows a diffusion law whose diffusion coefficient is increased with the flow maximal velocity $V$. Note that the prefactor is larger here than for a unidirectional flow of similar velocity (Appendix 2).

This theoretical prediction is confirmed by numerical simulations, comparing the long-range transport of particles of diameter 40 nm (in the range of commonly observed embryonic extracellular secreted vesicles in the CSF [*Bachy et al., 2008*], without flow, and in the presence of a bidirectional flow (*Figure 6B*) of variable velocity. The simulations show that at times longer than the characteristic diffusion time $d^2/D$ = 10 s, a diffusive-like gaussian concentration profile is recovered with a wider extent in the presence of a flow (*Figure 6B*). This diffusive transport is numerically obtained for three different maximal speeds of the flow: 0, 6 µm/s, and 20 µm/s (*Figure 6C1*), and shows that this effective diffusion is enhanced by the flow speed, as predicted by our theoretical derivation. Another interesting feature of this diffusivity enhancement is its weaker dependence on the size of the transported particles, as opposed to classical diffusion for which the diffusivity is inversely proportional to the particle size (*Figure 6C2*, *Equation 7*).

In vivo front tracking experiments illustrate this flow-induced accelerated transport of 20 nm nanoparticles in paralyzed and contracting zebrafish embryos (*Figure 6D*). While both contracting and paralyzed embryos exhibit a diffusive-like transport, the effective diffusivity is higher in the case of contracting embryos. It confirms that muscle contractions can increase the long-range transport efficiency (an increase of the steady bidirectional velocity would have similar effects, *Figure 6C1*). Interestingly, the transport is predicted to be enhanced in both rostro-caudal and caudo-rostral directions, which is confirmed by in vivo experiments. When injected locally in the caudal part of the CC, fluorescent nanobeads are indeed transported rostrally towards the brain ventricles (*Figure 6E1*). Similarly to rostral-to-caudal transport, the reverse transport is sped up incontracting embryos (*Figure 6E2*).

## Embryonic CSF circulates between the brain ventricles and the central canal via a complex circulatory system

To understand how the CSF carries particles between CC and the brain ventricles at early stages, we imaged all the CSF-filled cavities by injecting the small dye Texas Red-Dextran (3,000 MW) in live 30 hpf embryos (*Figure 7*). At this stage, cilia in the brain ventricles are mostly non-motile (*Olstad et al., 2018*). At this stage, CSF flow in the brain ventricles is mostly Brownian, although the heart beat is responsible for a pulsatile component to the flow at the heart beat frequency (*Video 8*), as previously observed (*Fame et al., 2016*; *Olstad et al., 2018*). Our observations indicate that before 30 hpf CSF flow in the brain ventricles does not contribute to the overall CSF circulation. After the brain ventricles, we investigated how CSF flows in between the brain ventricles and the CC. During embryogenesis in rats (*Sevc et al., 2009*) and zebrafish (*Kondrychyn et al., 2013*,

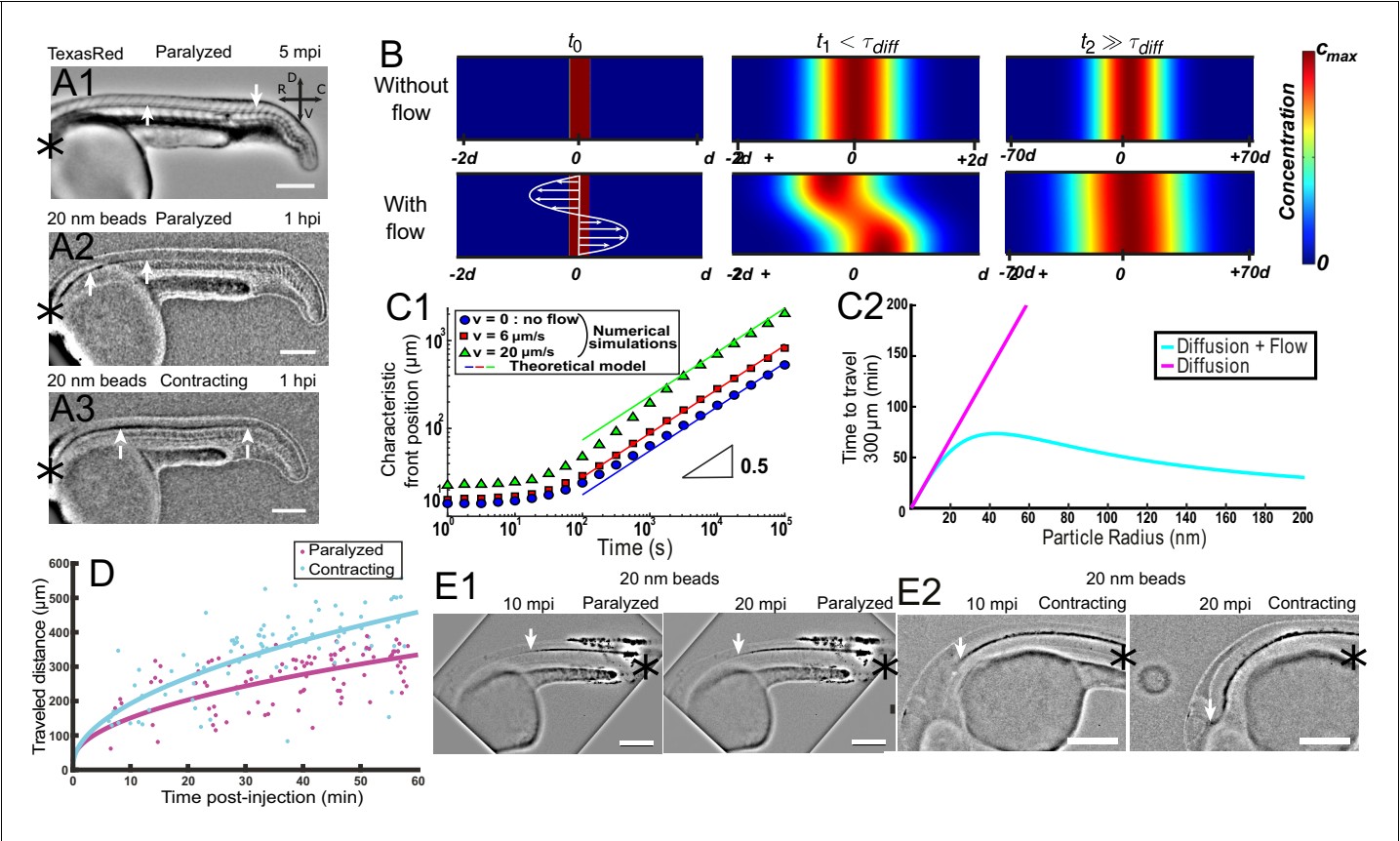

**Figure 6.** Local CSF flow accelerates bidirectional long-range transport along rostro-caudal axis in 30 hpf zebrafish embryos. (A1) Inverted contrast fluorescence widefield image of the propagation front of Texas Red 5 min post-injection (mpi). (A2, A3) Inverted contrast fluorescence widefield image of the propagation front of 20 nm beads 1 hr post-injection in the DV/MV of a paralyzed (A2) and contracting (A3) embryo. White arrows in (A1, A2, A3) show CC and the propagation front. (B) Numerical simulation of Taylor Aris diffusion, accounting for long-range transport in central canal. A small region of high concentration is created at the beginning of the simulation (t0 – left column) with (bottom) and without (top) the flow. The propagation front first follows the local flow profile (t1, center column), but a Gaussian distribution is finally recovered with and without flow (t2, right column). However, the concentration front has propagated faster in the presence of flow. The color represents the local concentration in particles. (C1) Characteristic front position of 20 nm radius particles versus time for different bidirectional flow velocities in log scale (v = 0, 6, and 20 $\mu m.s^{-1}$ for respectively the blue dots, the red boxes, and the green triangles), as calculated numerically. The colored solid lines correspond to the asymptotic prediction (0.5 line) of pure diffusion. (C2) Theoretical calculation of the average propagation time of a solution of spherical particles to travel 300 $\mu m$ in central canal versus the particle radius with (cyan) and without (magenta) the bidirectional flow. (D) Experimental propagation front of injected 20 nm beads versus post-injection time, measured in 16 paralyzed (magenta) and 16 contracting (cyan) embryos. (E1, E2) Inverted contrast fluorescence widefield image of the propagation front of 20 nm beads injected in the caudal central canal of a paralyzed (E1) and active zebrafish embryo (E2), showing 'reverse' (caudal to rostral) transport between left (10 mpi) and right images (20 mpi). The black asterisks in (A1–A3, E1, E2) show the injection site. Scale bar is 200 $\mu m$ in (A1, A2, A3, E1, E2).

*Ribeiro et al., 2017*; *Guo et al., 2018*), the progressive closing of the neural plate gives rise to a 'primitive lumen'(pL) that changes shape and volume over time. At 30 hpf in zebrafish embryos, we investigated here three transient paths where CSF flows in between the brain ventricles and the CC (*Figure 7*). First, the CSF circulates in a small ventral opening of the RV pL, in continuity with the ventral larger opening of the the CC pL. (as shown in histology in *Fame et al., 2016*). We measured that the diameter of this ventral opening is smaller than the CC diameter (7.1 ± 0.7 $\mu m$) and is 233 ± 24 $\mu m$ long (*Figure 7A and B* – green arrow, 7C,11 embryos, *Video 9*) and that few motile cilia enable there a unidirectional flow from rostral to caudal, that is towards the CC (*Video 10*). This fast and unidirectional flow follows a parabolic Poiseuille-like velocity profile with a maximum speed of 15 $\mu m$ / s (*Figure 7D1 and D2*). We also observed that CSF circulates in a ventrally-located small opening of the DV/MV pL connected with the TV (*Figure 7B*- red arrow, *Video 9*). Finally, we observed a small dorsal opening of the CC pL with a very narrow diameter (~2.5 $\mu m$, *Figure 7A and*

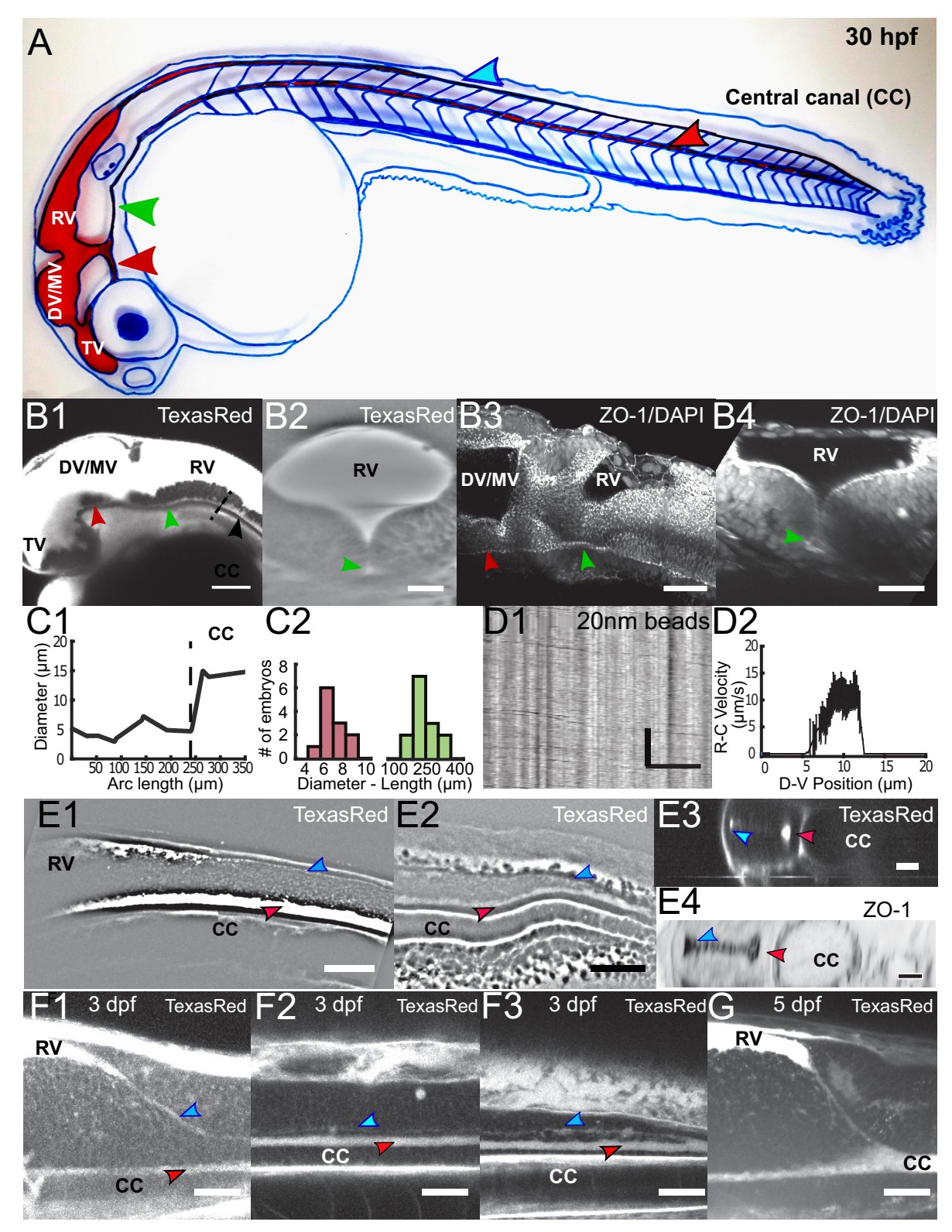

**Figure 7.** Embryonic CSF circulates between the brain ventricles and the central canal in the spinal cord through a complex circulation system. (**A**) Handmade schematic of CSF distribution in a 30 hpf zebrafish embryo with CSF colored in red. A path ventral to the DV/MV and the RV lets CSF flow and merges into the CC. Dorsal to the CC, a path that is continuous to the RV merges with the CC in the filum terminale. (**B1, B2**) 2-photon imaging of TexasRed injected in the DV/MV of 30 hpf embryos shows the ventral opening of the primitive lumen of DV/MV and RV in lateral (**B1**) and coronal (**B2**)
*Figure 7 continued on next page*

*Figure 7 continued*

views. The path ventral to RV merges into the CC and has a smaller diameter. (**B3, B4**) two photon imaging of tight junction (ZO-1) and nuclei (DAPI) labeling of fixed 30 hpf embryos in lateral (**B3**) and coronal (**B4**) views highlights the complex circulation system of CSF in primitive lumens. (**C1**) Experimental measurement of the ventral opening of the RV lumen relative to its arc length. The diameter sharply increases at the beginning of the CC (black dashed line). (**C2**) Histogram of the ventral opening of the RV lumen showing diameter and length measured in 11 WT 30 hpf embryos (diameter of $7.1 \pm 0.7$ μm, ratio relative to CC diameter of $0.55 \pm 0.12$, as discussed in Methods, and $233 \pm 24$ μm long median $\pm$ 0.5 s.d.). (**D1**) Inverse contrast kymograph of 20 nm beads flowing in the path ventral to RV lumen after injection in the DV/MV. (**D2**) CSF flow profile inside the path ventral to RV, captured with our automatic kymograph analysis. (**E1, E2**) 2-photon imaging of TexasRed injected in the DV/MV of a 30 hpf embryo showing the dorsal opening of the CC primitive lumen as it leaves the RV (**E1**) and as it connects to the main opening of the CC lumen in the filum terminale (**E2**). The dorsal opening of the CC primitive lumen in its rostral part is also shown in coronal view (**E3**) and in a fixed 30 hpf embryo after a ZO-1 staining (**E4**). (**F1, F2, F3**) 2-photon imaging of TexasRed injected in the DV/MV of a three dpf larva showing the CC lumen in rostral (**F1**), central (**F2**), and caudal positions (**F3**). (**F4**) 2-photon imaging of TexasRed injected in the DV/MV of a five dpf larva indicates that the caudal end of the RV is at this stage directly connected to the single opening of the CC. Scale bar is 100 μm in (**B1**), 50 μm in (**B3, E1, E2**), 30 μm in (**B2, B4, F1, F2, F3, G**), and 15 μm in (**D1, E3, E4**). The vertical scale bar in (**D1**) is 3 s.

**E** –blue arrow, *Video 11*). The CC pL is indeed composed of two main openings, with a very narrow but continuous connection between the two (*Kondrychyn et al., 2013*, *Ribeiro et al., 2017*). We observed that the apical domains of the two openings of the CC pL, marked with the tight junctions protein ZO-1, are closely-spaced, so that CSF flow is independent in each opening (*Figure 7E*). This is the reason why we neglected the influence of this dorsal opening of the CC pL to model the CSF flow in the larger opening, and we called the CC only its larger ventral opening. Interestingly, we found that the CC ventral and dorsal openings converge into a single opening, dense in motile cilia, in the caudal part of the spinal cord (*Figure 7 E2*, *Video 12*). During development, as it has already been partly described in rodents and zebrafish (*Sevc et al., 2009*; *Kondrychyn et al., 2013*, *Ribeiro et al., 2017*), dorsal and ventral openings of the CC pL converge ventrally at three dpf (*Figure 7F*) to finally merge into a single central canal lumen at five dpf (*Figure 7G*). Further histological investigations will be needed to clarify the geometry of all cavities where CSF flows in the embryo.

## The connection between the brain ventricles and central canal is critical for embryonic growth

In previous sections, we showed that long-range transport of particles in the CSF is accelerated by a fast and unidirectional flow between the ventricles and the CC, and by a slower but efficient bidirectional flow in the CC. We now investigate whether particle transport between the brain and the spinal cord is important for the development, or whether the ventricular and spinal CSF systems are independent at this stage. We performed 2-photon mediated photo-ablations at the beginning of the CC to disconnect the CC from the brain ventricles (*Figure 8A1 and A2*). We found that 30 hr after ablation, half of the ablated larvae were shorter in length than siblings that underwent a control ablation of the yolk extension or had no ablation (*Figure 8B1 and B2*). Larvae with the opening ablated had an increase in body-axis curvature (*Figure 8C1 and C2*). Similar effects on development were observed when CSF from the brain ventricles was drained (*Chang et al., 2016*), which affected the transport of CSF, including RBP4, a transport protein involved in the synthesis of Vitamin A. These experiments indicate that global CSF circulation contributes to embryonic growth, and suggest that the transport of particles between brain and spinal cord is a critical process for development.

In order to find particles that could be efficiently transported by the bidirectional flow, we investigated the content of the CSF and identified abundant extracellular vesicles using the label-free imaging technique, full field optical coherence tomography (*Figure 8D* and

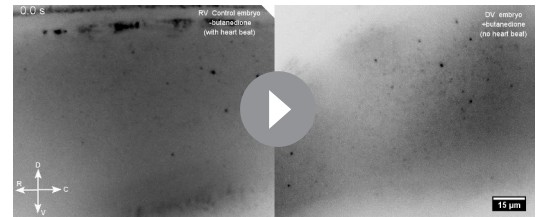

**Video 8.** 10 Hz video of 20 nm beads in the rhombencephalic ventricle of two 30 hpf zebrafish embryos with (left panel) and without (right panel) heartbeat with spinning disk microscopy. The heart was stopped using butanedione. The video is replayed at 10 fps.
https://elifesciences.org/articles/47699#video8

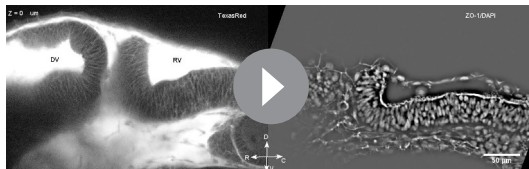

**Video 9.** 2-photon 3D imaging of CSF-filled primitive lumens around the brain of a 30 hpf zebrafish embryo. The left part shows the diencephalic/mesencephalic ventricle (DV/MV), rhombencephalic ventricle (RV) lumens with their respective ventral openings filled with TexasRed injected in the DV/MV. The same structures can be observed in a fixed embryo immunolabeled with ZO-1 (tight junctions) and DAPI at the right part of the movie.

https://elifesciences.org/articles/47699#video9

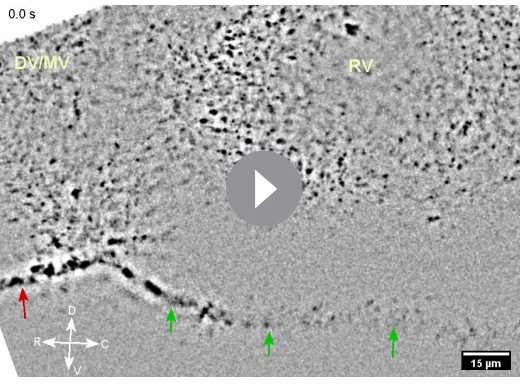

**Video 10.** 10 Hz video of 20 nm beads entering the ventral opening of the RV lumen of a 30 hpf zebrafish embryo with spinning disk microscopy. The video is replayed at 35 fps.

https://elifesciences.org/articles/47699#video10

*Video 13*). Numerous vesicles with high lipids or proteins content filled the CC, in agreement with previous observations (*Sternberg et al., 2018*). Fluorescent labeling of CD63 (*Video 14*, Materials and methods) and electron microscopy with immunogold labeling of exosomes suggested that some of these particles may correspond to exosomes in the CSF (*Figure 8E*, Materials and methods) (*Verweij et al., 2019*). Electron microscopy also reveals a variety of other extracellular vesicles, such as lipoparticles, microvesicles. The diameter of these vesicles ranges between 50 and 250 nm (*Figure 8E*), typically in a regime where Taylor-Aris dispersion is important. The bidirectional flow of endogenous extracellular vesicles can be put in evidence in the CC using full field optical coherence tomography as well as fluorescence live imaging (*Video 13*, *Video 14*). Altogether, our works suggests that the effective CSF circulation contributes to embryonic growth via the transport of extracellular vesicles between brain and spinal cord.

## Discussion

In this study, we performed quantification of cerebrospinal fluid (CSF) circulation in zebrafish embryos. By combining experiments and modeling, we demonstrated that a steady bidirectional CSF flow is driven by ventral motile cilia and that body movements create additional transient flows. These two driving forces accelerate the long-range transport of all particles larger than 10 nm, including the many macromolecules and extracellular vesicles present in the CC. This permits fast transport both from brain to spinal cord, and from spinal cord to the brain, which is critical for normal embryonic development. Our study focused on embryogenesis as this is a critical period when CSF is most concentrated in secreted extracellular lipid vesicles (*Bachy et al., 2008*; *Tietje et al., 2014*), and when blood flow along the body is not established to transport key molecules.

Our work predicts the establishment of a positive pressure in the CC, increasing linearly towards the caudal direction (*Figure 3—figure supplement 1A*). This inner pressure is decreased as the density of motile cilia is lower. These findings may be directly related to the reduced diameter of the CC reported for mutants with defective cilia (*Figure 4C*), an interesting observation reported for the first time

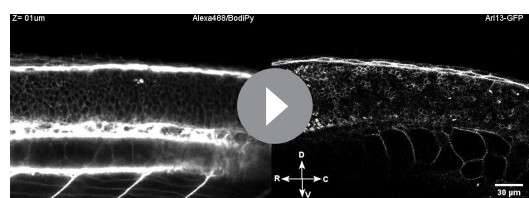

**Video 11.** 2-photon 3D imaging of CSF-filled structures in the spinal cord of a 30 hpf zebrafish embryo. The left part shows the lumen of the central canal (CC), with its large opening and a smaller dorsal opening (blue arrow). The two openings can be observed at the right part of the movie in a live 30 hpf *Tg(beta-actin:Arl13b-GFP)* embryo with labeled cilia, as characterized by a higher density of cilia.

https://elifesciences.org/articles/47699#video11

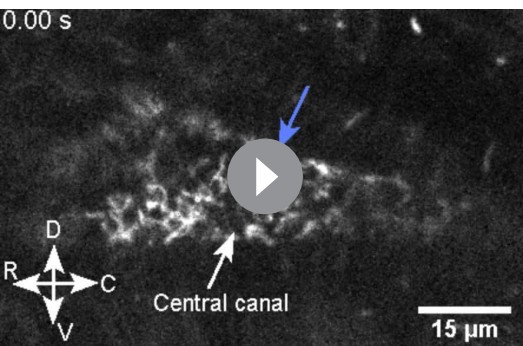

**Video 12.** 100 Hz video of the fork where the two openings of the CC primitive lumen merge in the filum terminale in the tail. Cilia are labeled with GFP in *Tg (β-actin: Arl13b-GFP)* transgenic embryos. The video is replayed at 20 fps.
https://elifesciences.org/articles/47699#video12

here. This suggests that, among other roles related to transport of chemical cues or mechano-sensing, the cilia motility ensures the integrity of the CC, which is necessary for proper body axis formation. Interestingly, small veins have also been reported to collapse under a decrease of inner pressure (*Fung, 2013*). During some fixation processes, if the structure is not maintained fast enough, the loss of cilia activity should lead to a decrease of the inner pressure and to the collapse of the CC. It may explain previous observations of collapsed CC in histology (*Orts-Del'Immagine et al., 2019*).We report that mutants with defective cilia were unable to maintain an open CC. *Elipsa* and *kurly* embryos with partial loss of cilia motility show intermediate phenotypes, with some regions open and other constrained, and with open regions showing bidirectional flow with a higher density of recirculation zones. These flow patterns are predicted by the sparse cilia model (*Figure 3C*), since a few cilia remain motile in *kurly* embryos (*Jaffe et al., 2016*) and in *elipsa* between 24 hpf and 30 hpf (*Cantaut-Belarif et al., 2018*). It also accounts for the reduced long-range transport previously observed in *kif3b* mutants, where cilia

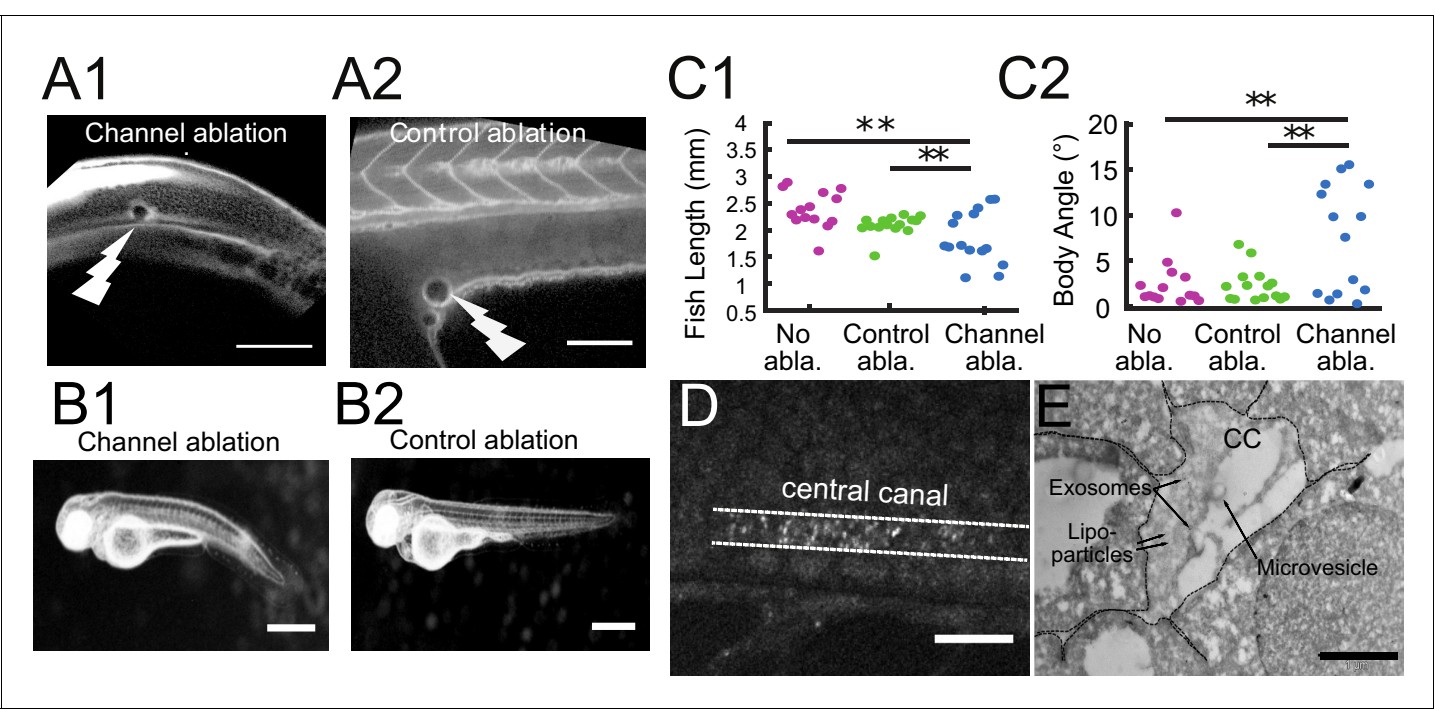

**Figure 8.** The connection between the brain ventricles and the central canal controls growth during embryogenesis. (**A1, A2**) 2-photon image of fluid-filled cavities after Texas Red injection and after optical ablation at the rostral extremity of the CC connecting to the brain ventricles (**A1**), and in the yolk extension (**A2** -control ablation). (**B1, B2**) Widefield transmitted images of 3 dpf zebrafish larvae 30 hr after the optical ablation of the CC rostral extremity (**B1**), or control ablation (**B2**). (**C1, C2**) Quantification of the fish total length (**C1**), and the main body angle (**C2**), 30 hr after Texas Red injection without ablation (magenta), after control (green) and CC rostral extremity l (blue) ablation with 14 larvae per category. P-values for length and angle are $1.3\ 10^{-3}$ and $3.0\ 10^{-3}$ respectively between the CC rostral extremity and no ablation, $9.6\ 10^{-3}$ and $6.4\ 10^{-3}$ between the CC rostral extremity and control ablation, and 0.066 and 0.89 (not significant) between the two controls. (**D**) Full field optical coherence tomography imaging of 30 hpf embryos reveals high density of secreted vesicles in central canal. (**E**) Electron microscopy with immunogold labeling of extracellular vesicles (EVs) in the CC of a one dpf zebrafish larva. Scale bar is 100 μm in (**A1, A2**), 15 μm in (**D**), and 500 nm in (**E**).

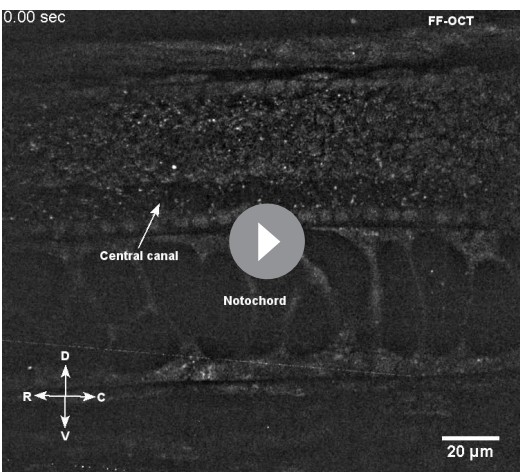

**Video 13.** 10 Hz video of lateral view of a 30 hpf zebrafish embryo with full field optical coherence tomography. The video is replayed at 15 fps.
https://elifesciences.org/articles/47699#video13

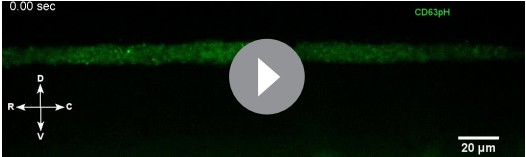

**Video 14.** Real-time video of the central canal of a one dpf zebrafish embryo expressing CD63-pHluorin in the YSL. The video is replayed at 35 fps.
https://elifesciences.org/articles/47699#video14

density is reduced (*Zhang et al., 2018*). In contrast, *foxj1a* mutant embryos showing ependymal cells with defective cilia (*Yu et al., 2008*) (*Figure 4—figure supplement 1*) have an almost closed CC. This observation strongly suggests that the motile cilia generating the flow and maintaining the CC geometry originate from ependymal cells only, while the remaining cilia that probably belong to CSF-contacting neurons (*Böhm et al., 2016*) are too short and not dense enough to generate a stable CC (*Figure 4—figure supplement 1*).

In this study, we finally described the global circulation of CSF through transient openings of the pL of the brain ventricles and the CC, and showed that a specific circulation between the ventricles and the CC is established at embryonic stage. We showed that this circulation was required for normal development. This transient architecture and circulation within the primitive lumens in the developing brain and spinal cord may be specific to teleost species, where the neural tube and ventricles form by tissue internalization and cavitation (*Guo et al., 2018*). In contrast, in Amniotes such as mammals, the neural tube forms mainly by tissue folding and fusion (*Araya et al., 2016*; *Cearns et al., 2016*; *Lowery and Sive, 2004*). However, there are evidences for similar appositions of ventricular walls in Amniotes (in mouse: *Chen et al., 2017* and; in chicken: *Halasi et al., 2012*), and these developing structures would be interesting to study in other species. In this study, we used a simple hydrodynamic model to account for our observations by modeling cilia contribution as a homogeneous volume force $f_v$. The latter fits experimental data for an amplitude of 4000 N/m$^3$, corresponding to a force of 10 pN generated by a single cilium occupying a volume of characteristic size 10 μm, in good agreement with the values reported in the literature (*Hill et al., 2010*). The force profile generated by the ciliary layer is most likely more complex spatially as well as temporally. Further refinements of the model may consist in changing the force profile $f_v$ for a linear static profile (*Siyahhan et al., 2014*), or more complex profiles (*Blake, 1972*; *Blake, 1973*; *Smith et al., 2007*). We report that the inclusion of a linear force profile does not significantly alter the predicted velocity profile (*Figure 3—figure supplement 1B*), which shows the robustness of this averaged model. Interestingly, our model can efficiently predict flow profiles for many other aspect ratios between the cilia extension and any other channel diameter and heterogenous ciliary beating frequency (*Figure 3—figure supplement 1C*). It convincingly predicts the flow previously reported by *Shields et al. (2010)* in the context of artificial magnetic cilia in microchannels , where this aspect ratio is low. However, we could not account for the complexity of many interacting cilia beating at very different frequencies, and it is only the steady average flow that could be predicted. We also neglected the contribution of dorsal cilia (although they represent about 20 % of all motile cilia), and we now provide further justifications for this choice. Exploiting the linearity of the Stokes equation, which prevails at these small scales, the CSF flow can be described as the sum of the contribution of ventral and dorsal motile cilia. The dorsal cilia, also oriented in the caudal direction, tend to cancel the contribution of their neighboring ventral cilia. Locally, beating dorsal cilia can thus abolish the bidirectional flow and create recirculation regions. This is analogous to the reported flow patterns in

mutants with decreased cilia activity. The bidirectional flow is thus affected by at most 20% due to the presence of sparse dorsal cilia.

The presence of the RF in the central canal was also neglected here, although it was proven to play a critical role in body axis straightening (*Cantaut-Belarif et al., 2018*). As the RF does not compartmentalize the fluid in 3D, its presence should not qualitatively modify the nature of the flow and its bidirectionality. This is confirmed by velocity profiles measured in *scospondin* mutants that lack a RF (*Cantaut-Belarif et al., 2018*). However, we expect a variation in the viscous stress in the vicinity of the fiber thereby changing the mechanical force applied by CSF flow on cells contacting the CC. As CSF-contacting neurons are known to be mechanosensitive (*Jalalvand et al., 2016*; *Sternberg et al., 2018*), a change of activity of these neurons in the absence of the RF may also occur. Our working model to account for the curved body axis of ciliary mutants is that the lack of motile cilia abolishes CSF flow, which leads to a reduction in CC diameter that alters RF polymerization (*Cantaut-Belarif et al., 2018*), and would alter the expression of URP peptides in CSF-contacting neurons (*Zhang et al., 2018*). The reason why RF would fail to form in a smaller CC remains an open question.

As a final thought, it might appear surprising that little is known about embryonic CSF flow today, but imaging this flow in vivo presents a daunting challenge. The CSF is a clear fluid with little endogenous contrast, and the CC is a small channel within the spinal cord that is difficult to access. In healthy young humans and other mammals in which the CC is open, the CC is often too small to be imaged with phase-contrast MRI (*Enzmann and Pelc, 1991*), unlike larger brain ventricles or subarachnoid spaces. Here, we take advantage of the transparency of zebrafish embryos to characterize CSF bidirectional flow and cilia dynamics in the CC. As the structure of the CC is very challenging to access with optical imaging in most species, but is well conserved among vertebrates (*Orts-Del'Immagine and Wyart, 2017*), we expect that our findings in zebrafish will guide future investigations in mammals. The thorough study of CSF flow and its role in larval, juvenile, and adult stages of life will be the object of future investigations. We expect the CSF flow to differ at later stages, as both the structure and the motility of cilia change over time in the CC.

Altogether, by combining models and quantitative data to demonstrate the origin and the role of the bidirectional flow, our study is a major step towards the understanding of CSF transport and its contribution to long-range communication between cells throughout the body that may well impact development, inflammatory response, and modulation of locomotor and postural functions carried by motor circuits.

# Materials and methods

## Key resources table

| Reagent type (species) or resource | Designation | Source or reference | Identifiers | Additional information |
|---|---|---|---|---|
| Genetic reagent (*D. rerio*) | hsc5Tg | (*Borovina et al., 2010*) | RRID: ZFIN_ZDB-ALT-100721-1 | referred to as Tg(β-actin: Arl13b-GFP) |
| Genetic reagent (*D. rerio*) | dnaaf1$^{tm317b/tm317b}$ | (*van Rooijen et al., 2008*) | ZDB-FISH-150901–13987 | referred to as lrrc50 |
| Genetic reagent (*D. rerio*) | foxj1a$^{nw2/nw2}$ | This paper | | Generated and characterized by N. Jurisch-Yaksi. See Material and methods, 'CRISPR/Cas9-mediated mutagenesis for foxj1a$^{nw2}$ mutants' |
| Genetic reagent (*D. rerio*) | traf3ip$^{tp49d}$ | (*Omori et al., 2008*) | RRID: ZFIN_ZDB-ALT-980413-466 | referred to as *elipsa* |
| Genetic reagent (*D. rerio*) | cfap298$^{tm304}$ | (*Brand et al., 1996*; *Jaffe et al., 2016*) | RRID: ZFIN_ZDB-ALT-980413-707 | referred to as *kurly* |

*Continued on next page*

*Continued*

| Reagent type (species) or resource | Designation | Source or reference | Identifiers | Additional information |
|---|---|---|---|---|
| Genetic reagent (*D. rerio*) | scospondin[icm15] | (*Cantaut-Belarif et al., 2018*) | RRID: ZFIN_ZDB-ALT-181113-4 | referred to as SCO |
| Antibody | mouse monoclonal IgG1 anti–ZO-1 | ThermoFisher | RRID: AB_2533147 | 1/200 Catalog # 33–9100 |
| Chemical compound, drug | FluoSpheresTM size kit #2, carboxylate-modified microspheres, yellow-green fluorescent (505/515), 2% solids, six sizes | Molecular Probes | F8888 | 20 nm beads were mostly used |
| Chemical compound, drug | TexasRed, dextran 3,000 MW, Neutral | ThermoFisher | D3329 | |
| Chemical compound, drug | α-Bungarotoxin | TOCRIS | 2133 | |
| Software, algorithm | Matlab | The MathWorks | RRID: SCR_001622 | |
| Software, algorithm | ImageJ | PMID: 22930834 | RRID: SCR_003070 | |
| Software, algorithm | COMSOL, Multiphysics | | RRID: SCR_014767 | |

## Animal husbandry

All procedures were performed on zebrafish embryos and larvae before 5 days post fertilization in accordance with the European Communities Council Directive (2010/63/EU) and French law (87/848) and approved by the Brain and Spine Institute (Institut du Cerveau et de la Moelle épinière). All experiments were performed on *Danio rerio* embryos of the AB strain, and were mitfa$^{-/-}$ (nacre) background. Animals were raised at 28.5℃ under a 14/10 light/dark cycle until the start of the experiment.

## Quantification and statistical analysis

All values are represented as median ± 0.5 standard deviation (s.d.) to evaluate the spread of each distribution, except for the error bars in the flow profiles showing the standard error of the mean (SEM), in which case we considered the average flow at one position. The speed distribution of particles depends on experimental errors and Brownian motion, and is therefore not considered. All statistics were performed using MATLAB. Statistical details related to sample size and p values, are reported in the figure legends. In figure panels, asterisks denote the statistical significance calculated by a two-tailed t test for samples with unequal variance (Welch t test): *, $p < 0.05$; **, $p < 0.01$; ***, $p < 0.001$; ns, $p > 0.05$.

## Injection in the CSF

30 hpf embryos were manually dechorionated, mounted in 1.5% low-melting point agarose in the lateral position. A few nanoliters of a mix containing 20 nm beads, 3,000 MW TexasRed dextran, and alpha-bungarotoxin (TOCRIS) were injected in the center of the diencephalic ventricle in order to label CSF and anaesthetize the fish in a single injection. The injections were performed using a Picospitzer device (World Precision Instruments). 20 nm carboxylate FluoSpheres of center wavelength 505/515 nm (yellow/green, F8888, Molecular Probes) were diluted to a 2% concentration in artificial CSF (containing in mM: 134 NaCl, 2.9 KCl, 1.2 MgCl2, 10 HEPES, 10 glucose, 2 CaCl2; 290 mOsM +/- 3 mOsm, pH adjusted to 7.8 with NaOH), and then sonicated for 10 s. 3,000 MW Texas Red dextran (Excitation/Emission wavelength: 595/615/ThermoFisher Scientific D3329) was diluted at 0.2% mass/volume concentration and was used both to serve as a marker of injection, and to

assess the boundaries of the ventricles and central canal. A final concentration of 100 μM of alpha-bungarotoxin was used. We could visually assess that such concentration, as well as the ventricle injection of alpha-bungarotoxin, although not conventional, were sufficient to paralyze the embryos. For experiments in contracting fish, the alpha-bungarotoxin was replaced by artificial CSF. Following the injection, we waited one hour before performing experiments to allow the particles to be transported down the central canal. Beads larger than 100 nm-diameter were restrained to brain ventricles, while beads of 20 and 45 nm diameter could enter the central canal.

## Fluorescent beads imaging

For CSF flow experiments, time-lapse images were acquired at room temperature on an inverted Leica DMI8 spinning disk confocal microscope equipped with a Hamamatsu Orca Flash 4.0 camera, using a 40X water immersion objective (N.A. = 0.8, pixel size 189 nm). 300 images of the beads mostly in the rostral part of central canal were acquired at a frame rate of 10 Hz for 30 s using Metamorph software (www.moleculardevices.com). Some experiments in contracting fish were performed at a frame rate of 40 Hz for 20 s to more accurately follow the muscle contraction.

## Particle tracking velocimetry

A MATLAB Particle Tracking Velocimetry (PTV) algorithm was adapted from a classical PTV algorithm (*Crocker and Grier, 1996*) and used to obtain instantaneous particle velocities. This algorithm was only used to produce *Figure 1B* as it only worked consistently in the best SNR conditions. The first crucial step was to segment the particles inside the region of interest corresponding to the central canal. This region was automatically inferred based on an intensity threshold on the low pass filtered mean image. Particle segmentation was more efficiently performed using a machine learning-based segmentation software (iLastik *Sommer et al., 2011*). Particle fine positioning was performed by fitting a 2D gaussian to each segmented region (particle) and by extracting the center position. A large matrix of particle position (X, Y, time) was obtained at this point. Then, the PTV algorithm itself was applied. The algorithm connects the particles found at successive times in order to minimize the total squared distance traveled by all beads. Note that this algorithm is optimized for purely Brownian displacements and limits the number of particles that one can track before saturating its computer memory, as the number of calculation scales as $N^2_{particles}$. Once the particles were tracked, each trace was obtained and kept only if it contained more than five time points. The instantaneous speed was calculated and plotted as a vector on the mean intensity image. An additional color code ranged from blue to red, corresponding to the minimal and maximal speed.

## Automatic kymograph measurement of beads velocity and CSF flow profile

First, time lapses were automatically rotated and cropped with an extra manual step for fine adjustments to isolate a portion of the central canal and place it in a correct dorso-ventral orientation. A 2D wavelet filter (decomposition of level six with Haar wavelet coupled to a 2-D denoising scheme based on level-dependent thresholds penalizing low signal pixels) was applied to increase the local signal-to-noise ratio of the fluorescent beads. Stacked images were then re-sliced and particle trajectories were automatically tracked and analyzed for every successive dorso-ventral position. The average signal on three consecutive positions was calculated with a moving-window average. Each column was then divided by its average value to account for spatially or temporally inhomogeneous light illumination and photobleaching if there was any. Only pixels above the average signal were kept, and an internal Matlab segmentation algorithm (*regionprops*) was applied to extract regions of interest in each kymograph. In order to filter only the straight lines corresponding to particle motion, a few additional empirical filtering conditions were applied: Only regions with at least 15 pixels (to avoid random fluctuations), and corresponding to lines with eccentricity above 0.9 (to prevent from selecting big aggregates or any large objects) were kept. Additionally, to prevent selecting particles stuck in the canal (vertical line in the kymograph), or camera/illumination fluctuations (global change of intensity in the entire image, leading to a horizontal line in the kymograph), we filtered any region whose orientation was close to 0 or 90°± 180° (excluding regions whose cosine or sine of orientation were below 0.1). Note that the intensity threshold was not strong, but most of the particle filtering was made through these four last conditions. A few regions corresponding to individual particle

rostro-caudal trajectory were thus obtained. The tangent of these line orientations was calculated, and corrected by pixel size and frame time, to obtain the individual particle velocity. All particles from one kymograph were aggregated to the average velocity and standard error to infer the local flow. Note that we refer to the number of events, or number of lines tracked, rather than the number of particles, as this algorithm cannot guarantee that one particle is not tracked several times if its dorsoventral position changes over time. The full MATLAB script can be found on Github. (*Thouvenin, 2019*; https://github.com/wyartlab/eLife_2019_OriginAndRole; copy archived at https://github.com/elifesciences-publications/eLife_2019_OriginAndRole).

## Extraction of parameters in the velocity profiles

We extracted some parameters from the flow profiles to compare the profile from many embryos. The maximal and minimal values, and their respective positions were extracted. The position at the center of the profile where the speed shifts from a positive to a negative value was also extracted. The position in the canal was normalized between 0 and 1, where 0 was the most ventral position where at least five moving particles were found and velocity exceeded 0.4, and one the most dorsal position where five moving particles were found and the velocity was not higher than −0.4.

## Flow versus time analysis

In *Figure 5D*, the average flow versus time was plotted in embryos able to contract in agarose. The experimental protocol was similar as for the flow profile analysis, except that alpha-bungarotoxin was removed from the injection mix. The same automatic kymograph approach was used to obtain events corresponding to motion of the beads. For each event, each trace was kept and its speed, its central position in D-V and R-C axis, and its time of occurrence were extracted. If the speed profile was obtained by averaging all pixels over time along the D-V axis, the average flow versus time was obtained by averaging all particles from all D-V positions at a given time. When bidirectional flow was present (before contraction, or in an anaesthetized embryo), the average flow was close to 0. in order for the flow measurement to be meaningful, and for the Brownian motion to be averaged out, several events should be measured at each timepoint. It gives a constraint on the actual timescale we can probe for the average flow rate. In *Figure 5D*, the fish contraction strength was estimated to be correlated to the temporary increase in volumetric flow. To estimate such contraction, the average signal from all pixels within central was calculated and the absolute value of their time derivative was calculated (magenta curve in *Figure 5D*). An arbitrary threshold of four was empirically chosen to select points during contraction. The contraction strength used in *Figure 5E* was calculated as the sum of the displacement signal during one event with displacements values above threshold. This value mixes the instantaneous derivative value and the contraction duration, as they can both play a role in the establishment of different flows. In datasets where embryos did not move, the average flow and displacement were calculated over the entire time series. The average displacement value was calculated in order to avoid getting a large value due to the large time span of the dataset and the use of absolute value.

## Comparison between methods to measure local flow (PTV/PIV/and kymograph approach)

Besides the PTV and kymograph approaches, we also tried to use a particle imaging velocimetry (PIV) algorithm, which is generally used to measure local flow (*Raffel et al., 2007*; *Santiago et al., 1998*). This section aims at comparing all these techniques qualitatively with respect to our imaging conditions. Before describing why we mainly used the automatic kymograph approach, we have to recall that we could not inject the particles directly in the central canal, and exogenous particles above 100 nm in diameter stayed confined inside the brain ventricles. Therefore, we could only image particles with diameter smaller than 100 nm, and mainly used 20 nm particles with fairly low signal, especially at high imaging frequencies. The Brownian motion of 20 nm beads is considerable and can generate instantaneous displacements of the same order of magnitude as the CSF flow itself. With that in mind, PIV should be efficient to measure the local flow even in low SNR conditions, as it uses a cross-correlation between successive frames. Nevertheless, because of the importance of Brownian motion, the cross-correlation fails at producing a peak of high SNR. Besides, because of the geometry of the canal and the bidirectional nature of CSF flow, we could not use a

large square region-of-interest to calculate the cross-correlation, as typically used in PIV. As particles can move in opposite directions in a $6 \times 6$ µm$^2$ ($32 \times 32$ pixels) square, the cross-correlation results in several small competing peaks. Asymmetric region of interests should be more adapted to take full advantage of the steady flow along the rostro-caudal direction, but performing a PIV with asymmetric regions is not standard and not trivial. On the other hand, PTV tracks every individual particle and should produce the trajectory of each particle versus time. One can expect that a given trajectory would then have a Brownian component plus a directed component, each of them being separable. Unfortunately, PTV was not very robust with low SNR, and not very efficient when particle density was quite high. The most critical part was to efficiently detect the particles without error. A first solution is to have a high threshold that selects only the brightest particles, as what was done in *Figure 1*. Nevertheless, only a few particles could be tracked for a given dataset, and the profile could not be inferred from only a few particle positions. If the threshold were lowered, it would increase the calculation time, and the probability of false detection. Since the PTV algorithm tends to minimize the distance traveled by particles, as the number of false detection increases, the false associations increased even more leading to an important underestimation of particle speed. In contrast, the automatic kymograph seemed more robust in this context because the flow is mostly steady in the rostro-caudal direction, allowing us to perform efficient averaging in this direction. Brownian motion was averaged out by averaging on 3–5 consecutive dorso-ventral positions, so that the particles produced traces long enough to be identified. The main strength of the kymograph approach was that many particles can be detected at every position, so that even if several mistakes were made, the standard error stayed small. The longer the total acquisition time, the more particles could be averaged, and the more precise the technique was, which was not true for both PTV and PIV in which all images were independent.

Nonetheless, most of our efforts focused on the automatic kymograph approach, and that probably PTV and PIV algorithms may be adapted to our configuration. We could not find any available PTV and PIV algorithm that could work with our data, and theories behind both techniques become more complicated in non-optimal configurations. In contrast, the kymograph just required to fit lines in an image, for which many solutions were available and just required small adjustments.

## Manual central canal diameter measurement

A Z-Stack of 30 images around the central canal of 30 hpf zebrafish embryos filled with Texas Red was acquired. A custom MATLAB program (*Thouvenin, 2019*, Github, https://github.com/wyartlab/eLife_2019_OriginAndRole) was used to open, rotate, and crop the image around central canal, then to calculate the maximum intensity projection. The rotation was crucial to fit the central canal projection by a manually selected rectangle, whose width measured the canal average diameter. Even though the canal diameter was not constant along the rostro-caudal direction, the rectangle was placed in order to minimize the total error. The canal diameter was averaged on large rostro-caudal portions, to provide fairly good estimate of its size. Data from wild type and mutant siblings were randomly mixed to perform blind analysis. For *Kurly* embryos displayed in *Figure 4*, Texas Red was not injected, but we used the time average image of fluorescent beads filling the central canal. The results on WT siblings showed remarkably similar measurements as those obtained with Texas Red. Thus, we decided to combine all these datasets.

## Live imaging of motile cilia in the central canal

30 hpf embryos from $Tg(\beta\text{-}actin{:}Arl13b\text{-}GFP;\ scospondin^{icm13/+})$ incrosses were manually dechorionated, laterally mounted in 1.5% low-melting point agarose, and paralyzed by injecting 1–2 nL of 500 µM alpha-bungarotoxin (TOCRIS) in the caudal muscles of the trunk. A Leica spinning disk confocal microscope equipped with a 40X water immersion objective (N.A. = 0.8) was used to measure cilia dynamics at a 100 frames per second for 3 s. Note that our estimation of beating frequency were limited to 50 Hz. Because we found that the average frequency of cilia beating in the central canal was higher than the value of 12–15 Hz previously reported (*Kramer-Zucker et al., 2005*), we imaged cilia dynamics with three different microscopes (two commercial spinning disk microscopes from 3i and Leica, and one 2-photon scanning microscope using the scanning artifact, as previously described by *Supatto and Vermot (2011)*, in order to validate our results). However, all the results displayed in this report were obtained with the same Leica commercial microscope to ensure data

homogeneity. We can only postulate that the difference found in beating frequency reflects a bias for long cilia with possible slower kinematics when using differential interference contrast (DIC) to identify cilia.

## Quantification of parameters related to cilia structure and dynamics

Datasets of cilia position over time were analyzed using custom MATLAB (*Thouvenin, 2019*, Github, https://github.com/wyartlab/eLife_2019_OriginAndRole) scripts based on Fourier transform. First, we opened, rotated, and cropped the image around the central canal. A straight line was manually drawn from the mean intensity image to define the center of the central canal. We also applied a 4 × 4 spatial sliding-average to increase the local signal-to-noise ratio. The temporal Fourier transform of each averaged pixel was calculated, and the position of its norm extracted and transformed into a frequency value. We could reconstruct 2D maps with the main frequency for each pixel (*Figure 2B2*). It drew distinct regions of interest of constant frequency, which we assumed to be the regions individual cilia could reach within one beating cycle. Within each region, the calculated main frequency can vary a few Hz, but was grouped around a central frequency with a narrow bandwidth. Successive mask images of pixels with main frequency within increasing bandwidth of 5 Hz (5–10 Hz, then 10–15 Hz, and so on until 50 Hz) were created, and analyzed using MATLAB image processing functions (regionprops). We selected as cilia only regions of constant frequency with area larger than 200 pixels (at least 3 μm in diameter). The position of each region was then compared to the position of the central line to label the cilia as dorsal or ventral cilia. First, the central frequency within each region was calculated. Because we extracted the average frequency over a large spatial scale, this measurement was independent of the cilium phase, and potentially free from sampling artifacts. Then, each region was associated to an ellipse, whose long axis was assumed to be the cilia length. After geometrical calculation, the ellipse orientation gave the angle between the cilia and the dorso-ventral axis. The values corresponding to the negative orientations may principally originate from the few beating dorsal cilia, also oriented towards the caudal end, but from dorsal to ventral (leading to a negative angle). To account for this, we calculated the median of the absolute value of the cilia orientation in the results section. Note that it gave the same value as the median of only the positive tilted cilia. Finally, we calculated a parameter called beating height, as the cilia length multiplied by the cosine of orientation (independent of the sign of the orientation), which gave an estimate of which portion of central canal was occupied by beating cilia. Although some reports (*Smith et al., 2012*) used the power spectral density to quantify the main beating frequency, we did not find a difference between the main frequency found with the Fourier transform versus the power spectral density calculation.

## CRISPR/Cas9-mediated mutagenesis for *foxj1a*$^{nw2}$ mutants

*foxj1a* mutants were generated with CRISPR/Cas9 mediated mutagenesis as previously described *Hwang et al. (2013)*; *Olstad et al. (2018)*. The gRNA sequence for *Foxj1a* gene (GGACGTGTGCTG TCCTGTGC) was identified using ZiFiT Targeter website (http://zifit.partners.org/ZiFiT_Cas9) from the genomic DNA sequence obtained from Ensembl. Annealed oligos were cloned into the pDR274 plasmid (Addgene Plasmid #42250). The sgRNA was in vitro transcribed using MAXIscript T7 kit (Invitrogen). Cas9 mRNA was in vitro transcribed from pMLM3613 plasmid (Addgene Plasmid #42251) using mMessage mMachine T7 kit (Invitrogen) and polyA tailed with a polyA tailing reaction kit (Invitrogen). A mixture of 25 pg of sgRNA and 600 pg of Cas9 mRNA were injected in embryos at one-cell stage. CRISPR/Cas9-induced mutations and germ line transmission of the mutations were detected by T7EI assay and characterized by sequencing. Three separate alleles were recovered from one founder. All mutations resulted in a frame shift from amino acid 66 and an early stop codon, which was before the forkhead domain. All homozygous mutants for the allele exhibited a bent body axis. This study used the *foxj1a*$^{nw2}$ allele, which carries a 5 bp deletion. See more information on *Figure 4—figure supplement 1*. Heterozygous adults and mutant larvae were genotyped using a KASP assay (LGC genomics) on a qPCR machine (ABI StepOne). The generation of the *Foxj1a* mutant line and fin clipping procedure on adults were approved by the Norwegian Food Safety Authorities (FOTS applications 16425 and 12395).

## Immunostaining and imaging of *foxj1a*$^{nw2}$ mutants

For immunostaining, straight body axis and curved body axis embryos resulting from an incross of heterozygous adults were selected and processed separately. The genotype of the clutch was confirmed by KASP genotyping of siblings presenting a curved body axis. Dechorionated and euthanized embryos (collected between 26 and 30 hpf) were fixed in a solution containing 4% paraformaldehyde solution and 1% DMSO for at least 1 hr at room temperature. Embryos were washed with 0.3% PBSTx (3 × 5 min), permeabilized with acetone (100% acetone, 10 min incubation at −20°C), washed with 0.3% PBSTx (3 × 10 min) and blocked in 0.1% BSA/0.3% PBSTx for 2 hr. Embryos were incubated with either glutamylated tubulin (GT335, 1:400, Adipogen) or acetylated tubulin (6-11B-1, 1:1000, Sigma) [MOU1] overnight at 4°C. The next day samples were washed (0.3% PBSTx, 3 × 1 hr) and incubated with the secondary antibody (Alexa-labeled GAM555 plus, Thermo Scientific, 1:1,000) overnight at 4°C. Next, the samples were incubated with 0.1% DAPI in 0.3% PBSTx (Life Technology, 2 hr), washed (0.3% PBSTx, 3 × 1 hr) and transferred to a series of increasing glycerol concentrations (25% and 50%). Stained larvae were stored in 50% glycerol at 4°C and imaged using a Zeiss Examiner Z1 confocal microscope with a 20x plan NA 0.8 objective.

## Central canal photodamage and photoablation

We intentionally damaged the central canal using different levels of light excitation in a few experiments. All experiments were performed after Texas Red injection. In experiments shown in *Figure 4D1*, a continuous medium intensity (25 mW) of green (565 nm) light was applied to the entire field of view 300 μm on the left and later 300 μm on the right of the displayed field of view for 5 min. After a few minutes of illumination, the central canal collapses, similarly to what was found in ciliary mutants (*Figure 4B2*). Debris were also seen, as shown on the sides in *Figure 4D1*. We never observed a breach in the central canal or fluorophores leaking out of it. A possible hypothesis is that TexasRed photobleaching due to the illumination may create damage to the central canal, cells surrounding it, or their motile cilia leading to an obstruction of the canal. In *Figure 8A1 and A2*, a light pulse of 200 mW (at a wavelength of 810 nm on a two-photon microscope) was scanned 200 times on 50 pixels lined along the dorso-ventral axis, right after the rostral extremity of the CC, or at the beginning of the yolk sac extension for control siblings. This reproducibly created a hole, and hence a breach in the central canal or in the yolk sac extension tissue, around which one can observe fluorescence leaking out in the surrounding tissue.

## Bead transport analysis

To measure CSF transport, four groups of eight 30 hpf embryos (16 contracting and 16 paralyzed embryos) were injected with beads and TexasRed within a five minutes time frame. We recorded the exact injection minute for each embryo. Widefield fluorescence images of each embryo were acquired every 10 min for 60 min using an epifluorescence microscope (Nikon AZ100) equipped with a 2X air objective imaged the entire embryo in one image. The image creation time recorded by the computer was automatically extracted and subtracted from the injection time. The front propagation was obtained by applying a low-pass filter on all images and by manually drawing the contour from the rostral extremity of the CC to the propagation front with a MATLAB script. The contour was drawn by successive lines changing direction every time the canal changed direction. The total length was calculated as the sum of all individual length. As for other manual arbitrary measurements, a random permutation was applied to the different images to reduce bias.

## Observation of CSF-filled cavities in 30 hpf embryos

We imaged CSF-filled cavitites with a 2-photon microscope (Vivo Multiphoton, 3i, Intelligent Imaging). Live embryos were injected with Texas Red and alpha-bungarotoxin in the diencephalic/mesencephalic ventricle. Other embryos were fixed in a solution containing 4% paraformaldehyde solution and 1% DMSO for at least 1 hr at room temperature. ZO-1 and DAPI immunostaining was then performed following a similar protocol as previously described. Z-stacks of between 30 and 300 μm (depending on the location) around either the brain ventricles or the central canal were obtained and displayed as a maximal projection to visualize the different lumens. *Video 9* was obtained by calculating the mean intensity every eight planes. Similar protocol was followed for 3 dpf and five dpf zebrafish larvae.

## Characterization of the ventral opening of the RV primitive lumen

30 hpf embryos were injected with Texas Red and alpha-bungarotoxin in the DV/MV and imaged with a confocal microscope (SP8, Leica). A Z-stack of 50 µm around the position of the ventral opening of the RV pL was obtained and displayed as a maximal projection to visualize its entire length as well as the beginning of central canal. A MATLAB script was used to perform measurement of this geometry. The contour of the ventral opening of the RV pL was manually drawn by drawing successive lines up to the most caudal part of the image for both the dorsal and ventral walls of this cavity. We could therefore measure the local diameter versus the curvilinear abscissa along the line. Its length was calculated between the initial point up to the point of maximal diameter derivative. Note that this method probably overestimated the lumen diameter, as proper alignment between the dorsal and ventral walls was not perfect. Nevertheless, the ratio of about two between the diameter of this ventral opening of the pL of the RV and the central canal should be in the right order of magnitude.

## Full-field optical coherence tomography imaging

The full-field optical coherence tomography (FF-OCT) setup was previously described (*Sternberg et al., 2018*; *Thouvenin et al., 2017*). The FF-OCT path is based on a Linnik interference microscope configuration illuminated by a temporally and spatially incoherent light source. A high power 660 nm LED (Thorlabs M660L3, spectral bandwidth 20 nm) provided illumination in a pseudo Köhler configuration. A 90:10 beam splitter separates the light into sample and reference arms. Each arm contains a 40x water immersion objective (Nikon CFI APO 40x water NIR objective, 0.8 NA, 3.5 mm working distance). In the reference arm, the light is focused onto a flat silicon wafer with a reflection coefficient of about 23.5% at the interface with water. FF-OCT detects and amplifies any structure that reflects or backscatters light within the sample arm, showing a contrast based on differences in refractive index, including small extracellular vesicles with rich lipid content. The beams from sample and reference interfere only if the optical path length difference between both arms remains within the coherence length of the system, ensuring efficient optical sectioning. A 25 cm focal length achromatic doublet focuses the light to a high speed and high full well capacity CMOS camera (Adimec, Q-2HFWC-CXP). The overall magnification of the FF-OCT path is 50x. The measured transverse and axial resolutions were 0.24 µm and 4 µm, respectively. Camera exposure was 9.8 ms, and images were acquired at 100 Hz. We acquired sequences of consecutive direct images and computed the standard deviation on groups of images to cancel the incoherent light that does not produce interference. Here, we also take advantage of phase fluctuations caused by the vesicle displacements occurring in the 10–100 Hz range to specifically amplify the signal while backscattered signal from the surrounding tissue is more static.

## Electron microscopy and secreted vesicles imaging

To induce expression of CD63-pHluorin, specifically labeling exosomes, embryos were injected at the 1000 cell stage. The following day, embryos were anesthetized with tricaine and embedded in 1.5% low melting point agarose. Live imaging recordings were performed at 28°C using a Nikon TSi spinning-wide (Yokagawa CSU-W1) microscope (*Verweij et al., 2019*). Other embryos were used for ultrathin cryosectioning and immunogold labeling, in which case they were fixed in 2% PFA, 0.2% glutaraldehyde in 0.1M phosphate buffer pH 7.4 at 3 dpf. Zebrafish were processed for ultracryomicrotomy and immunogold labeled against GFP using PAG 10. All samples were examined with a FEI Tecnai Spirit electron microscope (FEI Company), and digital acquisitions were made with a numeric camera (Quemesa; Soft Imaging System) (*Verweij et al., 2019*).

## Computational Fluid Dynamics (CFD) and numerical methods for solving the equations

We used MATLAB to obtain the numerical resolution of the system of three coupled equations detailed in the Results subsection 'A bidirectional flow generated by the asymmetric distribution of beating cilia'. Our simulations were performed by the finite element method solver COMSOL, using the Creeping Flow Module. The flow simulations were carried out by solving the Stokes equation in a rectangular domain composed of 120088 domain elements and 15046 boundary elements (corresponding to 630516 degrees of freedom). The simulations accounting for the transport of particles

in the presence of a flow were performed by coupling the diffusion equation to the Stokes equation in the same rectangular domain, using the Transport of Diluted Species Module. Doubling the number of mesh elements did not affect the results obtained throughout this study, and all the simulations converged.

## Data availability

Key datasets and custom MATLAB scripts used in this study can be found on Dryad server (14-04-2019-RA-eLife-47699) and on Github (*Thouvenin, 2019*, Github, https://github.com/wyartlab/eLife_2019_OriginAndRole).

## Acknowledgements

We thank Prof. Rebecca Burdine for the *kurly* (cfap*tm304* allele) mutant, Prof. Jarema Malicki for the *elipsa* mutant (*ift54/traf3ip*) and Prof. Brian Ciruna for the *Tg(beta-actin:Arl13b-GFP)* line. We would like to acknowledge the kind and expert assistance of Aymeric Millecamps, Tudor Manoliu, and Basile Gurchenkov from the ICM. Quant imaging facility for instrument use and scientific assistance on optical imaging, as well as Sophie Nunes-Figueiredo, Bogdan Buzurin, and Monica Dicu from PHENOZFish platform for fish care. We are grateful to Martin Catala for advising us on zebrafish histology. We greatly thank Jenna Sternberg for article proofreading. We thank the Yaksi laboratory members and the fish facility support team for scientific and technical assistance to NJK. This work was funded by Human Frontier Science Program (HFSP) Research Grant (grant n° RGP063-2018), and the New York Stem Cell Foundation (NYSCF) (grant n° NYSCF-R-NI39) and an ICM postdoctoral fellowship kindly attributed to OT. The research leading to these results has also received funding from the program 'Investissements d'avenir' ANR-10-IAIHU-06 (Big Brain Theory ICM Program), ANR-11-INBS-0011 (NeurATRIS: Translational Research Infrastructure for Biotherapies in Neurosciences), and a Helse Midt-Norge Samarbeisorganet grant to NJY and the Kavli Institute for Systems Neuroscience at NTNU.

## Additional information

### Competing interests

Claire Wyart: Reviewing editor, *eLife*. The other authors declare that no competing interests exist.

### Funding

| Funder | Grant reference number | Author |
|---|---|---|
| Human Frontier Science Program | RGP00063/2018 | Francois Gallaire<br>Claire Wyart |
| NIH Blueprint for Neuroscience Research | U19NS104653 | Martin Carbo-Tano<br>Claire Wyart |
| European Research Council | 311673 | Yasmine Cantaut-Belarif<br>Martin Carbo-Tano<br>Claire Wyart |
| New York Stem Cell Foundation | NYSCF-R-NI39 | Claire Wyart |
| Institut du Cerveau et de la Moelle Epinière | | Olivier Thouvenin |

The funders had no role in study design, data collection and interpretation, or the decision to submit the work for publication.

### Author contributions

Olivier Thouvenin, Conceptualization, Data curation, Software, Formal analysis, Investigation, Visualization, Methodology; Ludovic Keiser, Conceptualization, Data curation, Software, Formal analysis, Validation, Visualization, Methodology; Yasmine Cantaut-Belarif, Martin Carbo-Tano, Frederik

Verweij, Nathalie Jurisch-Yaksi, Data curation; Pierre-Luc Bardet, Project administration; Guillaume van Niel, Supervision, Investigation; Francois Gallaire, Supervision, Funding acquisition, Investigation, Project administration; Claire Wyart, Conceptualization, Supervision, Funding acquisition, Validation, Investigation, Methodology

**Author ORCIDs**
Olivier Thouvenin (iD) https://orcid.org/0000-0003-4853-7555
Nathalie Jurisch-Yaksi (iD) http://orcid.org/0000-0002-8767-6120
Claire Wyart (iD) https://orcid.org/0000-0002-1668-4975

**Ethics**
Animal experimentation: All procedures were performed on zebrafish embryos before 2 days post fertilization in accordance with the European Communities Council Directive (2010/63/EU) and French law (87/848) and approved by the Brain and Spine Institute (Institut du Cerveau et de la Moelle épinière).

**Decision letter and Author response**
Decision letter https://doi.org/10.7554/eLife.47699.sa1
Author response https://doi.org/10.7554/eLife.47699.sa2

## Additional files

**Supplementary files**
• Transparent reporting form

**Data availability**
The data enabling to plot all graphs for figures and supplemental videos have been deposited to Dryad Digital Repository doi:10.5061/dryad.4mj3pv1. The full MATLAB script can be found on Github https://github.com/wyartlab/eLife_2019_OriginAndRole (copy archived at https://github.com/elifesciences-publications/eLife_2019_OriginAndRole).

The following dataset was generated:

| Author(s) | Year | Dataset title | Dataset URL | Database and Identifier |
|---|---|---|---|---|
| Wyart C | 2019 | Data from: Origin of bidirectionality of cerebrospinal fluid flow and impact on long range transport between brain and spinal cord | http://doi.org/10.5061/dryad.4mj3pv1 | Dryad Digital Repository, 10.5061/dryad.4mj3pv1 |

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

## Appendix 1

## 2D model for the cilia-driven bidirectional flow in the central canal

We here provide a more detailed derivation of the flow velocity profile in a model 2D central canal, with a diameter $d$ and a ciliary region of height $h$. In the ciliary region (also called ventral region), the Stokes equation can be written:

$$\frac{dP}{dx} - f_v = \mu \frac{d^2 v_{ventral}}{dy^2} \tag{A1}$$

while in the non-ciliary region (also called dorsal region), the Stokes equation does not contain the bulk force $f_v$, but only the common pressure gradient:

$$\frac{dP}{dx} = \mu \frac{d^2 v_{dorsal}}{dy^2} \tag{A2}$$

Integration of **Equation A1** between $y_A$ and $y$ leads to:

$$\frac{dP}{dx}(y - y_A) - f_v (y - y_A) = \mu \frac{dv_{ventral}}{dy} \tag{A3}$$

where $y = y_A$ corresponds to the $y$-position for which the shear $dv/dy$ is zero. $y_A$ is a constant that will be derived in the following. Similar integration in the dorsal region [B] yields:

$$\frac{dP}{dx}(y - y_B) = \mu \frac{dv_{dorsal}}{dy} \tag{A4}$$

where $y_B$ is an analogous constant representing the $y$-position where a null shear is obtained.
In the ventral region [A], an additional integration of **Equation A3**, with the no-slip boundary condition ($v = 0$) at $y = 0$ (ventral wall) provides:

$$\frac{dP}{dx}\left(y^2/2 - y_A y\right) - f_v \left(y^2/2 - y_A y\right) = \mu v_{ventral}(y) \tag{A5}$$

Similarly in the region [B], with the no-slip b. c. ($v = 0$) at $y = d$, we get from **Equation A4**:

$$\frac{dP}{dx}\left(\frac{y^2}{2} - y_B y - \frac{d^2}{2} + y_B d\right) = \mu v_{dorsal}(y) \tag{A6}$$

In both **Equations (A5) and (A6)**, three unknowns remain to be determined in order to get the velocity profile: $dP/dx$, $y_A$ and $y_B$. At the border between the regions [A] and [B] ($y = h$), the continuity of velocity ($v_A(y = h) = v_B(y = h)$) and shear ($dv_A/dy(y = h) = dv_B/dy(y = h)$) can be written respectively:

$$-\left(\frac{dP}{dx} - f_v\right) y_A h - \frac{f_v h^2}{2} = \frac{dP}{dx}\left(-y_B h - \frac{d^2}{2} + y_B d\right) \tag{A7}$$

and

$$-\left(\frac{dP}{dx} - f_v\right) y_A - f_v h = -\frac{dP}{dx} y_B \tag{A8}$$

The last required equation to solve the system is provided by the condition of null global flux across a section of the channel, which is expressed in this bidimensional geometry:

$$\int_0^d v(y) dy = 0 \tag{A9}$$

that is:

$$\int_0^h v_{ventral}(y)\,dy = -\int_h^d v_{dorsal}(y)\,dy \tag{A10}$$

which finally provides:

$$\frac{dP}{dx}\left(h^3/6 - y_A h^2/2\right) - f_v\left(h^3/6 - y_A h^2/2\right) = \\ \frac{dP}{dx}\left(-d^3/3 + y_B d^2/2 - h^3/2 + y_B h^2/2 + hd^2/2 - y_B dh\right) \tag{A11}$$

The system of three **Equations A7, A8 and A11** contains three unknowns $dP/dx$, $y_A$ and $y_B$, and may be solved numerically. In the specific case of a ciliary layer occupying half of the central canal diameter ($l = d/2$), the solution is simple and symmetrical: $dP/dx = f_v/2$, $y_A = d/4$ and $y_B = 3d/4$ which leads to this symmetrical velocity profile:

$$v_{ventral} = \frac{dP/dx - f_v}{\mu}\left(y^2/2 - yd/4\right) \tag{A12}$$

$$v_{dorsal} = \frac{dP/dx}{\mu}\left(y^2/2 - 3yd/4 + d^2/4\right) \tag{A13}$$

## Appendix 2

## 2D model for the diffusive-convective transport of solutes in the central canal with a bidirectional flow

We here provide further details concerning the derivation of the effective diffusivity $D_{eff}$ of particles in the central canal in presence of a bidirectional flow.

In the CC, the motile cilia occupy the ventral half (**Figure 2**). From the flow model derived in Appendix 1 (**Equations (A3) and (A4)** of the main manuscript), one can rewrite the velocity profile changing the y-coordinate origin (y = 0 at the center of the channel) and using the maximal velocity $V_{max}$ = 6 µm/s:

$$v(y) = 16V_{max}\left(\frac{y}{2d} - \frac{y^2}{d^2}\right) \tag{A14}$$

in the dorsal region (positive values of y), and:

$$v(y) = 16V_{max}\left(\frac{y}{2d} + \frac{y^2}{d^2}\right) \tag{A15}$$

in the ventral region (negative values of y). The full transport equation of particles of local concentration c writes:

$$\partial_t c \;+\; v\partial_x c = D\partial_y^2 c + D\partial_x^2 c, \tag{A16}$$

where v(y) corresponds to the two velocity profiles written previously (**Equations (A14) and (A15)**), as the local velocity field is uniquely oriented in the rostro-caudal x-direction (see also **Figure 3** of the main manuscript). D is the molecular diffusion coefficient, here taken as $D = 10^{-11}$ m²/s, corresponding to particles of diameter 40 nm using the Stokes-Einstein law (**Equation A7** in the main manuscript). Several terms of **Equation (A16)** can be neglected considering their order of magnitude. With L the length of the channel, and d its diameter, the **Equation (A16)** may be written in dimensional analysis for times of order $T' L^2/D = \tau_{diff}$ (characteristic spanwise diffusion time):

$$\frac{c}{\tau_{diff}} + \frac{V_{max}c}{L} = D\frac{c}{d^2} + D\frac{c}{L^2} \tag{A17}$$

$$D\frac{c}{L^2} + \frac{V_{max}c}{L} = D\frac{c}{d^2} + D\frac{c}{L^2}, \tag{A18}$$

or

$$1 + \frac{V_{max}L}{D} = \frac{L^2}{d^2} + 1 \tag{A19}$$

Both $V_{max}L/D$ and $L^2/d^2$ terms are much larger than unity, due to the high aspect ratio of the channel L/d. This allows to simplify **Equation (A16)** in the two dorsal (y > 0) and ventral (y < 0) regions, respectively, as follows:

$$16V_{max}\left(\frac{y}{2d} - \frac{y^2}{d^2}\right)\partial_x c = D\partial_y^2 c, \tag{A20}$$

and

$$16V_{max}\left(\frac{y}{2d} + \frac{y^2}{d^2}\right)\partial_x c = D\partial_y^2 c. \tag{A21}$$

Adapting the derivation of **Bruus (2008)**, that was done for a Poiseuille flow in a cylinder, we first assume that the axial derivative of the concentration $\partial_x c$ is independent on y. This assumption will be verified a posteriori and confronted to numerical simulations.

Consequently, *Equations (A20) and (A21)* are ordinary differential equations for $c(y)$, that can be solved. A first integration between $y$ and $y = d/2$ for *Equation (A20)* for $y > 0$ leads to

$$16\frac{V_{max}}{d^2}\left(\frac{dy^2}{4} - \frac{y^3}{3} - \frac{d^3}{16} + \frac{d^3}{24}\right)\partial_x c = D(\partial_y c - \partial_y c|_{y=d/2}) \tag{A22}$$

Similarly, integrating *Equation (A21)* between $y = -d/2$ and $y$ for $y < 0$ leads to

$$16\frac{V_{max}}{d^2}\left(\frac{dy^2}{4} + \frac{y^3}{3} - \frac{d^3}{16} + \frac{d^3}{24}\right)\partial_x c = D(\partial_y c - \partial_y c|_{y=-d/2}). \tag{A23}$$

By antisymmetry, $\partial_y c|_{y=d/2} = -\partial_y c|_{y=-d/2}$. The equality of the concentration $y$-derivative in $y = 0$ then leads to $\partial_y c|_{y=d/2} = -\partial_y c|_{y=-d/2} = 0$, which simplifies *Equations (A22) and (A23)*. A second integration between $y = 0$ and $y$ of *Equation (A22)* for $y > 0$ leads to

$$\frac{4V_{max}d^2}{3D}\left(\frac{y^3}{d^3} - \frac{y^4}{d^4} - \frac{y}{4d}\right)\partial_x c = (c(y) - c(0)). \tag{A24}$$

Similarly, a second integration between $y = 0$ and $y$ of the *Equation (A22)* for $y < 0$ leads to

$$\frac{4V_{max}d^2}{3D}\left(\frac{y^3}{d^3} + \frac{y^4}{d^4} - \frac{y}{4d}\right)\partial_x c = (c(y) - c(0)). \tag{A25}$$

We now aim at expressing the concentration profile with respect to the average concentration $\bar{c}$ over a cross section

$$\bar{c} = \frac{1}{d}\int_{-d/2}^{d/2} c(y)dy, \tag{A26}$$

which, using *Equations (A24) and (A25)*, yields

$$\bar{c} = \frac{4V_{max}d\partial_x c}{3D}\left(\int_{-d/2}^{0}\left(\frac{y^3}{d^3} + \frac{y^4}{d^4} - \frac{y}{4d}\right)dy + \int_{0}^{d/2}\left(\frac{y^3}{d^3} - \frac{y^4}{d^4} - \frac{y}{4d}\right)dy\right) + c(0) \tag{A27}$$

and finally

$$c(0) = \bar{c}. \tag{A28}$$

This theoretical prediction is confirmed by numerical simulations solving the complete transport equation without the assumptions made for the theoretical model (*Appendix 2—figure 1*), for different times, positions in the canal and flow velocities.

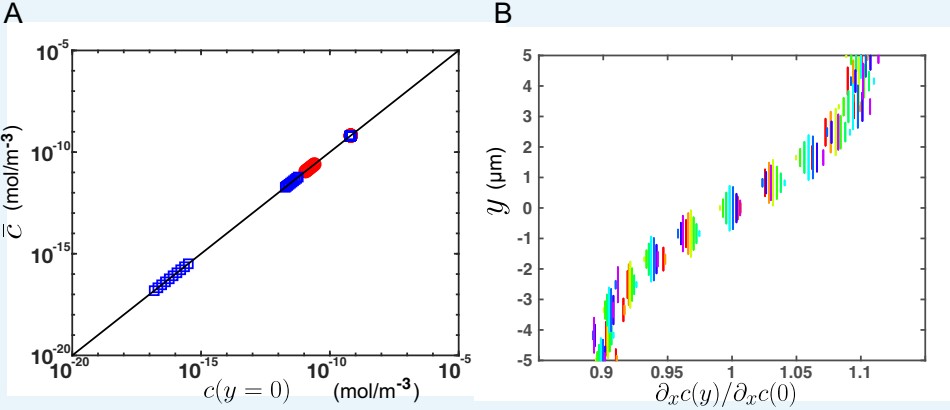

**Appendix 2—figure 1.** Additional results from numerical simulations supporting the theoretical

model for transport. (**A**) Average concentration over a cross section versus concentration at the center of the CC (y = 0), for different flow velocities (Vmax = 6 or 20 μm/s), at different times and for different x-position in the channel, obtained from numerical simulations (FEM, COMSOL) on a 2-D central canal. The black line corresponds to $\bar{c} = c(0)$ and confirms the predictions of the theoretical model. (**B**) Relative variation of the axial gradient of concentration $\partial_x c$ over a cross section (y-position), at various x-positions (corresponding to different colors), from numerical simulations. $\partial_x c$ varies of about 10% around its mean value over a cross section, which justifies the assumptions done in the model (from **Equations (A7) and (A8)** to **Equations (A9) and (A10)**) to neglect its variations in comparison with the strong variations of the other integrand $\left( \frac{y^3}{d^3} + \frac{y^4}{d^4} - \frac{y}{4d} \right)$.

Going back to our theoretical model, one can now derive the condition for having a y-independent axial gradient of concentration (condition required for recovering a diffusive-like concentration profile) by differentiating the **Equations (A24) and (A25)** with respect to x. The latter condition is consequently fulfilled only if $\partial_x c = \partial_x c(0)$, which requires the third term to be negligible. In terms of order of magnitude, this implies that

$$\frac{4V_{max}d^2}{3D}\frac{1}{L} \ll 1, \tag{A29}$$

where L is the length of the central canal. This condition can be rewritten as a condition on the Péclet number Pe:

$$Pe = \frac{V_{max}d}{D} \ll \frac{3L}{4d} \simeq 100. \tag{A30}$$

This condition on the Péclet number is always fulfilled in the CC of zebrafishes for nanobeads or lipidic vesicles ($D \sim 10^{-11}$ m²/s), where Pe does not exceed 10.

We now aim at deriving the effective diffusivity of particles in the CC. We calculate the average current density J(x) through the cross-section, using the velocity profiles (**Equations (A14) and (A15)**) and the concentration profiles (**Equations (A24) and (A25)**):

$$J(\bar{x}) = \frac{1}{d} \int_{-d/2}^{d/2} \rho c(x,y) v(x,y) dy. \tag{A31}$$

The zero average flux over the cross section leads to the vanishing of the constant component $\bar{c}$ of **Equations (A24) and (A25)**. The previous expression may then be simplified:

$$J(\bar{x}) = \frac{64V_{max}^2 d}{3D} \left( \int_{-d/2}^{0} \left( \frac{y^3}{d^3} + \frac{y^4}{d^4} - \frac{y}{4d} \right) \left( \frac{y}{2d} + \frac{y^2}{d^2} \right) dy \right.$$
$$\left. + \int_{0}^{d/2} \left( \frac{y^3}{d^3} - \frac{y^4}{d^4} - \frac{y}{4d} \right) \left( \frac{y}{2d} - \frac{y^2}{d^2} \right) dy \right) \rho \partial_x \bar{c}, \tag{A32}$$

$$J(\bar{x}) = \frac{64V_{max}^2 d}{3D} \left( \int_{-d/2}^{0} \left( -\frac{y^2}{8d^2} - \frac{y^3}{4d^3} + \frac{y^4}{2d^4} + \frac{3y^5}{2d^5} + \frac{y^6}{d^6} \right) dy \right.$$
$$\left. + \int_{0}^{d/2} \left( -\frac{y^2}{8d^2} + \frac{y^3}{4d^3} + \frac{y^4}{2d^4} - \frac{3y^5}{2d^5} + \frac{y^6}{d^6} \right) dy \right) \rho \partial_x \bar{c}, \tag{A33}$$

which may be approximated to

$$J(\bar{x}) \simeq -\frac{V_{max}^2 d^2}{24D} \rho \partial_x \bar{c}. \tag{A34}$$

The effective diffusivity $D_{eff}$ may then be directly derived from the effective Fick's law

$$D_{eff} \simeq \frac{V_{max}^2 d^2}{24D} = \frac{Pe^2}{24} D. \tag{A35}$$

A more detailed calculation in a cylindrical channel was done by **Aris (1956)**. It generalizes this result at intermediary Péclet numbers Pe ~ 1. Adapted to our geometry and flow, it yields

$$D_{eff} \simeq D\left(1 + \frac{Pe^2}{24}\right) \qquad (A36)$$

For particles of radius 20 nm (corresponding to the size of injected nanobeads), we have $D \sim 5 \cdot 10^{-12}$ m²/s, which leads for $V_{max} = 6$ µm/s to an effective diffusivity $D_{eff} \sim 2.5D$. Similarly, the effective diffusivity of the same particles for a faster flow of amplitude $V_{max} = 20$ µm/s is $D_{eff}' 18D$. These two theoretical predictions are in good agreement with the numerical simulations (**Figure 6C** of the main manuscript).

One strong assumption of the model that enabled to readily integrate **Equations (A7) and (A8)** considers that the axial derivative of the concentration $\partial_x c$ is only weakly varying over a cross section, such that one can assume it is constant. The **Appendix 2—figure 1** plots the relative variation of the normalized concentration gradient $\partial_x c / \partial_x c$ (0) as a function of y. It shows that this quantity exhibits minor variations of about 10% over a cross section, demonstrating the relevance of the assumption in the derivation.

