## [Decision Letter]

**Acceptance summary:**

How fluid flow and cilia motility modulate robust and reproducible patterning events in the embryo is of great interest to a large audience of scientists, both in developmental biology and more disease oriented disciplines. Cilia motility controls the embryonic cerebrospinal fluid circulating between the brain ventricles and the central canal in the spinal cord, failure of which could lead to congenital disease. This work studying cerebrospinal fluid dynamics in the living zebrafish embryo, exploiting imaging, genetics and modelling approaches, sheds new light on a particular mechanical integration between cilia motility and fluid movements within the cerebrospinal canal. Of particular interest is that deficiency in fluid movement can impairs robust morphogenesis and embryonic growth, raising the possibility that similar defects may compromise similar events in human. The precise effectors of cilia motility and the anatomy of the systems of canal where fluids flow will require future work. This study will certainly stimulate others in the field to build on the authors’ working model.

**Decision letter after peer review:**

Thank you for submitting your article "Origin of bidirectionality of cerebrospinal fluid flow and impact on long range transport between brain and spinal cord" for consideration by *eLife*. Your article has been reviewed by two peer reviewers, and the evaluation has been overseen by a Reviewing Editor and Didier Stainier as the Senior Editor. The following individuals involved in review of your submission have agreed to reveal their identity: Andrej Vilfan (Reviewer #1).

The reviewers have discussed the reviews with one another and the Reviewing Editor has drafted this decision to help you prepare a revised submission.

Summary:

This is a highly interesting study of fluid flow in the central canal. The paper first presents in vivo maps of the cilia and the flow, then provides a simple model for the bidirectional velocity distribution, showing that the cilia-induced pressure difference helps maintaining the shape of the canal and discussing the implications of the bi-directional flow for particle transport (which has the properties of enhanced diffusion) and embryogenesis. The study describes an automated workflow of bidirectional CSF flow assessment in the central canal (CC) of the zebrafish spinal cord as well as of CC geometry and cilia localization, which the authors can closely mimic in their modeling attempts. Mutant lines with defects in ciliary motility are analyzed using this workflow, demonstrating a reduced CC diameter (a collapse) in mutants. The manuscript concludes with the description of a novel, so-far overlooked, anatomical feature of the zebrafish central nervous system (Figure 7): while the central canal should run all the way from the ventral diencephalon to the caudal end of the spinal cord and is connected to the diencephalon by a newly discovered funnel cranially, the rhombencephalon extends a novel, much thinner canal dorsally which connects to the CC caudally by another thin canal. If this claim can be substantiated, this finding definitely warrants publication in *eLife*.

Essential revisions:

1) The existence of the newly described channel, that – if true – needs to be named, has to be proven beyond reasonable doubt, which requires in depth histology, ideally at TEM resolution. The brain anatomy in Figure 7A and Video 8 are not clear, particularly the identity and connection of brain ventricles to CC and the new canal, which is actually not displayed except as a thin line in the scheme in Figure 7A. In other vertebrates (amphibians through mammals), the diencephalon harbors a blind-ending ventral protrusion of the ventricle, which will give rise to the neural part of the hypophysis. Do we see flow in the hypophysis protrusion of the diencephalon in Video 8? Would the hypophysis then be connected to the CC? Why should the rhombencephalon not be connected to the CC, which is what we know from all other vertebrates? This might be an ancestral feature of basal vertebrates, which would be extremely interesting in an evolutionary context, but it needs to be proven. Known canal-like connections in vertebrate embryos include the buccohypophyseal (ventral diencephalon) and neurenteric canals (caudal neural tube/spinal cord). How do the described canals relate to the new one?

2) It is very interesting to see that in ciliary motility mutants the CC collapses and CC flow is abolished. The authors speculate about potential function of CC flow and exosomes transported along the CC. Based on the mutant phenotypes, what specific function of CC flow can be derived? One would really like to learn more about the physiological function of CC flow. A careful analysis/discussion of spinal cord phenotypes in mutants should provide the/an answer.

3) The authors argue that 30 hpf is the ideal time point for their analysis. With the improved automated workflow at hand, the descriptive part of this work would gain a lot by relating this stage to earlier and later time points, particularly with respect to the novel canal.

4) Is it possible to know more about the few motile dorsal cilia: how many are they? Are they polarized, i.e. would they contribute to bi-directional flow? Where are they found along the cranial-caudal axis?

5) Figure 2B, Figure 2—figure supplement 1:

The zig-zag pattern of some cilia in Figure 2—figure supplement 1A illustrates some challenges when inferring the ciliary beating frequency from a single pixel intensity, which is a highly non-linear function of the cilium's phase. In the middle of the zig-zag line, the principal peak in the power spectrum will likely correspond to the second harmonic of the beating frequency. Aliasing can be an additional problem (the 2nd harmonic of a cilium beating with 40 Hz appears at 20 Hz when sampled with 100 fps). The description in Materials and methods is unclear on how these problems were dealt with.

6) Equation (4): the force density *f_v_*changed its sign between Equations (2) and (4).

7) Equation (5): The zero flux condition (5) must have been taken into account in the derivation of Equations (3) and (4), which always give v=0 at y=d/2. Yet it is only introduced afterwards.

In the subsequent text, the value of α is given, but not that of the viscosity mu. The pressure gradient, which is half the value of *f_v_*, is presented as a numerical result, rather than a basic symmetry property of the model.

8) "Counterintuitively, this model predicts that vortices may originate not from cilia dynamics, but rather from the local absence of motile cilia in the ventral side on a distance larger than *d*."

What is the basis for this statement? What is meant by "cilia dynamics"? It is clear that cilia and gaps between them are both needed to get recirculation.

Moreover, the solution shows an interesting mirror symmetry about the center of the channel, which is not discussed. But it is easy to understand, considering that the velocity profile remains unchanged if any force profile f(y,z)=f(z) (i.e., independent of y) is added to the channel. The force distribution in Figure 3 can therefore always be made antisymmetric with respect to the channel center.

9) Figure 7: the funnel and the loop through RV are very difficult to see – it would be helpful to redesign the figure and make a magnification of the relevant region.

[Editors' note: further revisions were suggested prior to acceptance, as described below.]

Thank you for resubmitting your work entitled "Origin and role of the cerebrospinal fluid bidirectional flow in the central canal" for further consideration by *eLife*. Your revised article has been evaluated by Didier Stainier (Senior Editor) and a Reviewing Editor.

The manuscript has been improved but there are some remaining issues that need to be addressed before acceptance, as outlined below:

The key point is to solve the general issue of the presumed canal system, especially given its transient nature. Some guidelines are provided below to address this issue.

Reviewer #2:

I like to repeat myself and state again that this manuscript by Wyart, Gallaire and colleagues is mainly technical and descriptive in nature (it lacks a hypothesis). It describes a novel, transient canal system in the embryonic zebrafish CNS; an automated workflow of bidirectional CSF flow assessment in the central canal (CC) of the zebrafish spinal cord as well as of CC geometry and cilia localization, which the authors can closely mimic in their modeling attempts. Several of the features of CC flow described here have been previously reported (by the same authors), though not at the resolution achieved here.

The novelty that qualifies this manuscript for *eLife* lies primarily in the description of the presumed novel channel system in the embryonic CNS. The revised manuscript contains a limited set of additional histological data that should prove the existence of these canals. Unfortunately, rather than being convinced, I now have doubts as to their existence.

The most parsimonious explanation for these canals would be that they represent aspects of the ventricle in contact with the floor plate ('tel- and diencephalospinal canals') and roof plate ('rhombencephalospinal canal') of the developing brain and spinal cord, with a temporal occlusion in between due to lateral walls getting in close contact (explaining why there is no fluid flow). In the chick embryo, Gary Schoenwolf has studied such a temporal occlusion back in the 1980s already. A wealth of published histological sections in various vertebrate embryos attest that brain ventricle walls as well as lateral walls of the spinal cord approach each other temporarily during development (cf. examples below). The transversal sections shown in the new Figure 7B2 and B4 can be interpreted in exactly that way. Zooming in on the DSC – or perhaps ventricle close to the floor plate – the tear-drop (with flattened bottom) shape of the presumed canal actually argues that this structure is not closed dorsally, which is visible in the ZO-1 stained specimen in B4, in which staining is continuous from the ventral-most aspect (DSC/FP) all along the lateral walls into the wide lumen of the rhombencephalon. The 'similar geometry' of Reissner's fiber and the ventral canal in addition argues that this canal represents the ventral most part of the ventricle next to the floor plate.

Because this issue is of central importance for the main message of this (descriptive) manuscript, it needs to be addressed beyond reasonable doubt: we have to be one hundred percent certain as to the nature of the described items. I think this is feasible without much effort and would include conventional histology, immunofluorescence staining and the analysis of some marker genes. I would suggest to stain WT specimens for mRNA expression of floor and roof plate marker genes (shh, BMP4, etc.) as well as for genes that highlight the dorso-ventral pattering of the brain and neural tube (pax6, nkx2.2 or the like). Stained specimens should be analyzed by conventional histology through transversal sections, such as in Figure 7B4. IF-staining for basal lamina-specific markers should be applied in addition to prove or disprove the existence of these canals.

If it turned out that the canals represented indeed merely the continuation of the ventricle lumen next to floor and roof plate, the manuscript would need to be rewritten. Whether or not it would still qualify for *eLife* is hard to judge at this point; it would need another round of thorough reviewing.

Example 1: from G. Halasi et al., Developmental Biology 365 (2012) 118-132;

Figure 1. Neural tube of WT chick embryo. Lumina at floor and roof plate are reminiscent of presumed canals in zebrafish.

Figure 2. Temporal progression of CC development, demonstrating the temporal nature of separate lumina at floor and roof plate.

Example 2: from Chen et al., Toxicol. Pathol. 45, 705-744, 2017. Figure 9. Representative images of the transient neural lumen occlusion of the spinal cord in the E11.5 mouse embryo.

[Editors' note: further revisions were suggested prior to acceptance, as described below.]

Thank you for resubmitting your work entitled "Origin and role of the cerebrospinal fluid bidirectional flow in the central canal" for further consideration by *eLife*. Your revised article has been evaluated by Didier Stainier (Senior Editor) and a Reviewing Editor.

The manuscript has been improved but there are some remaining issues that need to be addressed before acceptance, as outlined below. The critical issues have been adequately addressed and the reviewers believe the work will receive considerable attention in the field. They however identified a few remaining points where simple text revisions will be required before acceptance, as outlined below. Overall, the authors are asked to eliminate the term canal and explain that further work is needed to clarify its geometry.

Reviewer #2:

Nature of the report:

As stated by the authors: we disagree on this point. Although the authors use state-of-the-art technology to visualize and model cilia and CSF flow in the developing zebrafish ventricular system, the main character of the study is descriptive; it lacks a conceptual advance, i.e. it remains unclear what the physiological meaning of bi-directionality is and how the specific pattern of flow relates to any ciliopathies.

Anatomy in question:

Unfortunately, the revised manuscript still lacks clarity. It now seems obvious that the structures the authors report – "conduction paths" (termed "canals" in the former versions of the manuscript, a term that persists in the revised manuscript as well) – represent the dorsal and ventral aspects of the developing ventricular system. This should be clearly stated in the manuscript, not only given as a possible explanation in the Discussion. Canals and conduction paths are confusing and imprecise; in addition, such terms tend to stick, which would be inappropriate. The cited paper by the Sive lab shows a parasagittal section, which does not resolve whether or not these structures are part of the ventricle and provides no proof for the existence of novel conduction paths. In addition, the Ribeiro et al. paper nicely shows that during transition of the neural tube primitive lumen into the central canal, the lumen becomes very narrow, but remains continuous between its dorsal and ventral aspects. If the authors insisted that their conduction paths and canals were independent entities, 3D imaging or morphing of histological sections and segmentation of the ventricular system would be required.

---

## [Author Response]

Essential revisions:1) The existence of the newly described channel, that – if true – needs to be named, has to be proven beyond reasonable doubt, which requires in depth histology, ideally at TEM resolution. The brain anatomy in Figure 7A and Video 8 are not clear, particularly the identity and connection of brain ventricles to CC and the new canal, which is actually not displayed except as a thin line in the scheme in Figure 7A. In other vertebrates (amphibians through mammals), the diencephalon harbors a blind-ending ventral protrusion of the ventricle, which will give rise to the neural part of the hypophysis.

We agree with the reviewer that the description of this “diencephalospinal” canal connecting the floor of the diencephalic ventricle at the level where it also connects to the rhombencephalic ventricle to the central canal (see revised Figure 7) required more investigation. Many small canals (2-5 microns in diameter) collapse when investigated in EM or histology. In order to elucidate the structure of small canals where the CSF circulates and small dyes can diffuse in zebrafish embryos, we find that the optimal approach is to image in living embryos the propagation of small fluorescent dyes. Without knowing the diffusion of small dyes in vivo, it is not clear to know where the CSF can flow. The complex 3D structure of the canals located ventrally from the ventricles would be difficult to capture in a whole embryo after fixation and slicing for histology, or even worse with TEM. The investigation of these channels with TEM and classical histology will be the focus of a future study.

We imaged CSF-filled structures by relying on TexasRed-Dextran 3,000 MW injected in vivoin the rhombencephalic ventricle and using optical sectioning from confocal and two photon laser scanning microscopies (see Materials and methods in revised manuscript). We also performed immunohistochemistry on ZO1 to label tight junctions along the CSF ventricles and canals, and compared our results to immunohistochemistry for DAPI to label nuclei. We drew a schematic summarizing our observations (Figure 7A), improved the illustrations in Figure 7 and added stacks in Videos 8 and 9.

Using this combination of approaches at the 30 hpf embryonic stage, we observed:

1) The telencephalospinal canal (see Figure 7A, Figure 7B, and new Video 8) previously referred to in the submitted manuscript as the “funnel”: this canal runs ventrally to the telencephalic, diencephalic and rhombencephalic ventricles, connecting telencephalic to diencephalic/mesencephalic ventricles, and then diencephalic/mesencephalic ventricle to central canal. The structural properties of this canal could only be quantified in the diencephalospinal portion. In this portion, the diencephalospinal canal differs from the central canal in terms of i) diameter (5 μm instead of 10 um) and ii) flow (unidirectional instead of bidirectional). Finally, note that this canal runs ventrally to the diencephalic ventricle and is therefore located several hundred microns more rostral than the rhomencephalon-spinal cord boundary where the central canal starts. These three arguments (diameter, flow and location) argue that the telencephalospinal canal is distinct from the central canal. Interestingly, the diencephalospinal canal is also the path where the Reissner Fiber forms from the aggregation of the *scospondin* by the subcommissural organ, the flexural organ and the floor plate (Cantaut-Belarif et al., 2018). Immunostaining for Reissner material shows a similar path in the diencephalospinal canal than what we described with fluorescent dyes, forming a curve before reaching the CC at 30 hpf, and straight in the larva after 48 hpf, as shown in Author response image 1.

**Author response image 1. respfig1:** Immunostaining for Reissner material in 30 hpf embryos (Top image) most likely forms through the diencephalospinal canal showing a similar geometry as this canal (TexasRed in living embryo – Bottom image).

The diencephalospinal canal is typically 7 μm in diameter and therefore too small for being easily identified after fixation in sagittal and transverse sections – although portion of this canal were already observed in Fame et al., Fluids and Barriers of the CNS 2016 – Figure 2B’, where fluorescein-labelled 2000 kDa dextran conjugated to fluorescein was injected in the ventricles of living 30 hours post fertilization (hpf) zebrafish embryo and the diencephalospinal portion of the canal was observed and classical histology with hematoxylin and eosin stain in Fame et al., Fluids and Barriers of the CNS 2016 – Figure 2C’’, showed over 50 μm the opening of diencephalospinal portion of the canal at the level. Our results i) corroborate these observations (see revised Figure 7B and 7C), ii) adds evidence for the anterior portion of the canal ventrally from telencephalic to diencephalic ventricle (Figure 7A, 7B), which was previously not described and iii) add the characterization of CSF flow in the posterior section of the canal (diencephalospinal canal) where 20 nm diameter fluorescent beads enter (Figure 7D). Note that this canal is now described and discussed with reference to the study of Fame et al., 2016 in the revised manuscript in a new section inside results named ‘The CSF circulates between the brain and the central canal inside the spinal through a complex system of channels at embryonic stage’.

2) A rhombencephalospinal canal (see revised Figure 7A, 7E and new Video 9) running dorsally from the rhombencephalic ventricle (RV) to the filum terminale in the caudal spinal cord where it can reach the central canal in the embryo. This canal could be named as well as an “accessory central canal”. This canal filled with fluorescent dyes is only 2.5 μm in diameter and runs for the entire length of the spinal cord in the dorsal spinal cord. At the larval stage 3 dpf and 5 dpf, we have evidence that this canal migrates ventrally (3 dpf, Figure 7F) to fuse with the central canal (5 dpf, Figure 7G). This observation corroborates previous images (Kondrychyn et al., 2013, Figure 3M to 3T). This canal present at the embryonic stage 30 hpf is to our knowledge reported here for the first time as an independent canal where CSF can flow. Nonetheless, two studies performing anti-ZO-1 immunostaining or DAPI staining (Kondrychyn et al., 2013; Ribeiro et al., Open Biology 2017) reported in transverse section a second lumen dorsal to the central canal in the embryonic stage. While these authors assumed that it could be a dorsal extension of central canal, our data indicates that it is a distinct canal as CSF does not flow along the dorso-ventral axis from the central canal to this canal, and as no CSF flow can be observed inside. We observe that the rhombencephalospinal canal connects to the RV rostrally and forms a fork when merging with the CC in the caudal end of the spinal cord (Figure 7E2, and 7F3, and new Video 10). This fork was also observed in living animals with cilia labeling in the *Tg(beta-actin:arl13-GFP)* (new Video 10).

Do we see flow in the hypophysis protrusion of the diencephalon in Video 8? Would the hypophysis then be connected to the CC?

The hypophysis indeed appears at this embryonic stage around the location where the telencephalospinal canal reaches the connection between diencephalic/mesencephalic ventricle and hindbrain ventricle (Pogoda and Hammerschmidt, Semin. Cell. Dev. Bio., 2007), though from images previously published it appears more ventral than the TSC at embryonic stage (Ning-Ai Liu et al., 2008, Dev. Bio). In our former Video 8 (now Video 11 in revised version), we did not observe flow in the brain ventricles at embryonic stage, except for vortexes observed where the diencephalospinal starts (Bend described in Author response image 1, and Video 12). Although we cannot rule out that this region is the infundibulum (the hypophysis’s protrusion), the precise location of the hypophysis would require an extensive study with molecular markers, which is out of the scope of our study.

Why should the rhombencephalon not be connected to the CC, which is what we know from all other vertebrates? This might be an ancestral feature of basal vertebrates, which would be extremely interesting in an evolutionary context, but it needs to be proven.

In zebrafish, the complex structure of multiple canals we describe, one running ventrally to the ventricles (TDS), one dorsalmost (RSC) and one (CC) ventral above the floor plate, is transient (Figure 7E, 7F, 7G). We observe these canals in the embryo at 30 hpf but it evolves by 5 dpf in the larva w here the rhombencephalospinal canal migrates ventrally towards the central canal in order to connect rhombencephalic ventricle to CC (Figure 7G – Fame et al., Fluids and Barriers of the CNS 2016, check Figure 2H), as observed in other vertebrates. We now illustrate these observations (Figure 7E, 7F, 7G) and discuss the transient nature of this structure in our revised manuscript and question whether similar mechanisms might occur in other Amniotes. In Anamniotes such as mammals where the neural tube and ventricles form in a very different manner (Massarwa et al., Development 2013; Massarwa et al., WIREs Dev Biol 2014), it is unlikely that the same transient architecture of canals form during development.

Known canal-like connections in vertebrate embryos include the buccohypophyseal (ventral diencephalon) and neurenteric canals (caudal neural tube/spinal cord). How do the described canals relate to the new one?

Our observations do not bring evidence that the canals we describe running ventrally to the telencephalic and diencephalic ventricles correspond to the buccohypophyseal canal (connecting to the mouth) nor the neurenteric canal (connecting to the intestine). Further characterization with molecular markers for these structures will be necessary in the future for validation.

2) It is very interesting to see that in ciliary motility mutants the CC collapses and CC flow is abolished. The authors speculate about potential function of CC flow and exosomes transported along the CC. Based on the mutant phenotypes, what specific function of CC flow can be derived? One would really like to learn more about the physiological function of CC flow. A careful analysis/discussion of spinal cord phenotypes in mutants should provide the/an answer.

The observation of a collapse of the central canal in mutants with defective cilia is novel and very interesting. We now added a paragraph in the Discussion of the revised manuscript:

“The observation of a collapse of the central canal in mutants with defective cilia is novel and very interesting. […] The mechanisms by which the collapse of CC and absence of RF contribute to the curled down phenotype will need to be further investigated.”

3) The authors argue that 30 hpf is the ideal time point for their analysis. With the improved automated workflow at hand, the descriptive part of this work would gain a lot by relating this stage to earlier and later time points, particularly with respect to the novel canal.

We agree with the reviewer that following the evolution of CSF flow during development is very interesting. We added elements to quantify our observations in embryos and larvae in Figure 7. A full quantification will however be the focus of a subsequent study as it would take us more than 6 months to properly complete.

4) Is it possible to know more about the few motile dorsal cilia: how many are they? Are they polarized, i.e. would they contribute to bi-directional flow? Where are they found along the cranial-caudal axis?

This is a great question. To answer points 4 and 5 of the reviewer, we performed novel live imaging experiments to quantify the properties of 1704 cilia in 40 embryos, and 89 fields of view. We improved our analysis workflow to classify automatically ventral versus dorsal motile cilia by drawing a line at the center of CC. Based on our analysis of ventral and dorsal cilia, the manuscript has been revised in the subsection Central canal geometry and properties of the motile cilia (Figure 2C see also Materials and methods ‘Quantification of parameters related to cilia structure and dynamics’) to describe dorsal cilia, as well as in the second section in Discussion to estimate how dorsal motile cilia can modify the local CSF flow.

We found that dorsal motile cilia represents 18% of all motile cilia and were evenly distributed along the rostro-caudal axis of CC. Dorsal cilia beat with a slightly smaller central frequency, but are otherwise quite similar to ventral cilia, with similar orientation and length. It indicates that the asymmetric distribution of motile cilia probably contributes the most to generate the bidirectional CSF flow, rather than a beating frequency difference.

We also discussed the contribution to the dorsal cilia to local CSF flow in the Discussion:

“We could estimate the effect of the dorsal cilia, which represent about 20% of all motile cilia (that we neglected in our model). […] The bidirectional flow is thus affected by at most 20% due to the presence of sparse dorsal cilia.”

As suggested by the sparse cilia model, we can also postulate that the local flow cancellation due to the presence of motile dorsal cilia is prone to generate areas of recirculation (vortices) where dorsal cilia are beating. We can also add that if the density of motile dorsal and ventral cilia were similar, it would completely cancel the CSF flow, as long as the flow rate keeps being 0.

5) Figure 2B, Figure 2—figure supplement 1:The zig-zag pattern of some cilia in Figure 2—figure supplement 1A illustrates some challenges when inferring the ciliary beating frequency from a single pixel intensity, which is a highly non-linear function of the cilium's phase. In the middle of the zig-zag line, the principal peak in the power spectrum will likely correspond to the second harmonic of the beating frequency. Aliasing can be an additional problem (the 2nd harmonic of a cilium beating with 40 Hz appears at 20 Hz when sampled with 100 fps). The description in Materials and methods is unclear on how these problems were dealt with.

We thank the reviewer for having pointed out this potential source of artifacts. In fact, spatial averaging was achieved before and after the Fourier transform, as a 4x4 sliding spatial average was applied, and as we select only large ROIs of constant frequency. We therefore believe our measurements are independent of the cilium’s phase. Because it was probably not clear enough in the initial version, we modified the description in Materials and methods of the revised manuscript:

“Datasets of cilia position over time were analyzed using custom MATLAB software based on Fourier transform. […] The position of each region was then compared to the position of the central line to label the cilia as dorsal or ventral cilia..…“

We have multiple arguments supporting the fact that we are not subject to aliasing or non linear artifacts in our experiments. First, we included now many more experiments performed at 100 Hz. By increasing the number of cilia imaged by a factor of 5, we did not observe changes in the distributions of frequency, length or angle. If we were sensitive to artifacts in a small percentage of cilia, increasing the number of cilia should have reduced the influence of artifactual data points on the distribution. Second, in experiments acquired at the different frequencies of 80 Hz and 120 Hz, the frequency distribution was similar to the one obtained at 100 Hz, whereas the presence of aliasing should have shifted the frequency distribution (towards 0 Hz or 40 Hz respectively). Finally, from the Fourier analysis, we could extract several peaks, and compared the frequency of the first peak versus the frequency of the second to check whether they were separated by a factor of 2, as it should be the case if the 2nd harmonic component were strong. Except for a few cases (less than 5% of cilia) the frequencies of the first two peaks were not related, making us more confident that our measured frequencies were accurate with a limited amount of 2nd harmonic measured.

6) Equation (4): the force density f_v_ changed its sign between Equations (2) and (4).

We thank the reviewer for this remark, as we made a typo in Equation (2). We modified accordingly such that a minus appears in front of the force *f_v_.*

7) Equation (5): The zero flux condition (5) must have been taken into account in the derivation of Equations (3) and (4), which always give v=0 at y=d/2. Yet it is only introduced afterwards.In the subsequent text, the value of α is given, but not that of the viscosity mu. The pressure gradient, which is half the value of f_v_, is presented as a numerical result, rather than a basic symmetry property of the model.

The reviewer is right, this final expression of the velocity profile already accounted for the zero flux expression. The general expression before the zero flux constraint is now written in the supplementary information, and exhibits a more complex form depending on the combination of dP/dz and *f_v_*. Applying the zero flux constraint, but for a general expression of the ciliary region thickness h, we obtain the new Equations (3) and (4), that exhibit a central velocity at y=d/2 which is not necessarily zero. In the particular case of the zebrafish embryo, we assumed that the force created by the cilia is applied to one half of the channel, as *h* is taken as *d/2*. In this case only the central velocity becomes zero.

8) "Counterintuitively, this model predicts that vortices may originate not from cilia dynamics, but rather from the local absence of motile cilia in the ventral side on a distance larger than d."What is the basis for this statement? What is meant by "cilia dynamics"? It is clear that cilia and gaps between them are both needed to get recirculation.

This sentence was indeed confusing and we apologize for this imprecision. Obviously, without cilia, i.e. without flow, there cannot be recirculation regions. The initial purpose was to distinguish the vorticity directly created by the cyclic 3D beating of a cilium, which our model cannot describe, and the less expected observed vortices created by the alternation of active and passive regions, that is a consequence of the confined geometry of the model central canal. As a consequence, our model predicts that the recirculation area should start not around to the active cilia, but where cilia are lacking. Note that this is consistent with the fact that the number of vortices increases in “intermediate” ciliary mutants (*kurly, elipsa*) that have a decreased density of motile cilia (but still have a few motile cilia). It was intriguing for us at first, because we expected the vortices to be created by the local beating of certain cilia, in which case we expected to have less vortices associated with a lower density of motile cilia. In the main text, we modified the above sentence by:

“This model thus predicts that vortices appear due to the alternation of active and passive regions, rather than due to the complex 3D beating of cilia. As a consequence, the vortices should not start around active cilia, but in regions with lower motile cilia density.”

Moreover, the solution shows an interesting mirror symmetry about the center of the channel, which is not discussed. But it is easy to understand, considering that the velocity profile remains unchanged if any force profile f(y,z)=f(z) (i.e., independent of y) is added to the channel. The force distribution in Figure 3 can therefore always be made antisymmetric with respect to the channel center.

It is a very interesting point that we have missed originally. We agree with the reviewer, yet only in the particular case of the force generated by cilia occupying one half of the channel, as is the case in the central canal of zebrafish. However our solution is made more universal (see the new Equations (3) and (4) where the universal *h* replaced the earlier *d/2*), accounting for arbitrary cilia-to-channel diameter aspect ratio, as illustrated in Supplementary Figure 1C. In the general case, this mirror symmetry disappears, which is why we decided not to develop further.

9) Figure 7: the funnel and the loop through RV are very difficult to see – it would be helpful to redesign the figure and make a magnification of the relevant region.

As mentioned earlier, Figure 7 was split into Figure 7 and Figure 8 in order to improve the number, quality and readability of the schematics.

[Editors' note: further revisions were suggested prior to acceptance, as described below.]

The key point is to solve the general issue of the presumed canal system, especially given its transient nature. Some guidelines are provided below to address this issue.Reviewer #2:I like to repeat myself and state again that this manuscript by Wyart, Gallaire and colleagues is mainly technical and descriptive in nature (it lacks a hypothesis). It describes a novel, transient canal system in the embryonic zebrafish CNS; an automated workflow of bidirectional CSF flow assessment in the central canal (CC) of the zebrafish spinal cord as well as of CC geometry and cilia localization, which the authors can closely mimic in their modeling attempts. Several of the features of CC flow described here have been previously reported (by the same authors), though not at the resolution achieved here.

There are two points raised here that we address separately:

A) On the novelty of our results: let us agree to disagree.

Yes, we previously analyzed manually in two studies with mutants the flow of CSF showing that it was bidirectional in the central canal during embryogenesis (Cantaut-Belarif et al., 2018; Sternberg et al., 2018).

However, we tackle in the study of Thouvenin et al., the mechanistic question of how CSF bidirectional flow is generated in the central canal and what can it be useful for during embryogenesis.

By combining original data-descriptive automated analysis of flow coupled with manipulations such as the ablations in Figure 4D1 and Video 5 – and thorough modeling, our article:

1) demonstrates that motile and polar cilia that are inserted on the ventral floor of the central canal are sufficient to sustain the bidirectional flow in the central canal;

2) shows original evidence that motile cilia, in addition to generate a bidirectional flow of CSF, influence the shape of the central canal;

3) shows how bidirectionality of CSF flow together with muscle contractions enhance longrange transport in the CSF.

None of these three important points have been reported before.

The novelty that qualifies this manuscript for eLife lies primarily in the description of the presumed novel channel system in the embryonic CNS. The revised manuscript contains a limited set of additional histological data that should prove the existence of these canals. Unfortunately, rather than being convinced, I now have doubts as to their existence.

No, this finding does not constitute the novelty of our study and no, these structures had been described using the diffusion of small molecules as well as histology in the publications we cited here:

– The structures are described in Fame et al., Fluids Barriers CNS 2016 as well as Kondrychin et al., Plos One 2013, and

– Data showing the same structures are visible but not commented on in Ribeiro et al., Open Biol. 2017 and Guo et al., iScience 2018.

The novelty of our study here lies in two points:

1) We characterize the flow in the diencephalospinal conduction path and show that in contrast to the central canal, the flow is there unidirectional;

2) We show by performing 2-photon mediated ablation at the connection between the diencephalospinal conduction path and the central canal that transport between the ventricles and the central canal impacts growth of the embryo.

The most parsimonious explanation for these canals would be that they represent aspects of the ventricle in contact with the floor plate ('tel- and diencephalospinal canals') and roof plate ('rhombencephalospinal canal') of the developing brain and spinal cord, with a temporal occlusion in between due to lateral walls getting in close contact (explaining why there is no fluid flow). In the chick embryo, Gary Schoenwolf has studied such a temporal occlusion back in the 1980s already. A wealth of published histological sections in various vertebrate embryos attest that brain ventricle walls as well as lateral walls of the spinal cord approach each other temporarily during development (cf. examples below). The transversal sections shown in the new Figure 7B2 and B4 can be interpreted in exactly that way. Zooming in on the DSC – or perhaps ventricle close to the floor plate – the tear-drop (with flattened bottom) shape of the presumed canal actually argues that this structure is not closed dorsally, which is visible in the ZO-1 stained specimen in B4, in which staining is continuous from the ventral-most aspect (DSC/FP) all along the lateral walls into the wide lumen of the rhombencephalon. The 'similar geometry' of Reissner's fiber and the ventral canal in addition argues that this canal represents the ventral most part of the ventricle next to the floor plate.Because this issue is of central importance for the main message of this (descriptive) manuscript, it needs to be addressed beyond reasonable doubt: we have to be one hundred percent certain as to the nature of the described items. I think this is feasible without much effort and would include conventional histology, immunofluorescence staining and the analysis of some marker genes. I would suggest to stain WT specimens for mRNA expression of floor and roof plate marker genes (shh, BMP4, etc.) as well as for genes that highlight the dorso-ventral pattering of the brain and neural tube (pax6, nkx2.2 or the like). Stained specimens should be analyzed by conventional histology through transversal sections, such as in Figure 7B4. IF-staining for basal lamina-specific markers should be applied in addition to prove or disprove the existence of these canals.If it turned out that the canals represented indeed merely the continuation of the ventricle lumen next to floor and roof plate, the manuscript would need to be rewritten. Whether or not it would still qualify for eLife is hard to judge at this point; it would need another round of thorough reviewing.Example 1: from G. Halasi et al. / Developmental Biology 365 (2012) 118-132;Figure 1. Neural tube of WT chick embryo. Lumina at floor and roof plate are reminiscent of presumed canals in zebrafish.Figure 2. Temporal progression of CC development, demonstrating the temporal nature of separate lumina at floor and roof plate.Example 2: from Chen et al., Toxicol. Pathol. 45, 705-744, 2017. Figure 9. Representative images of the transient neural lumen occlusion of the spinal cord in the E11.5 mouse embryo.

B) On the nature of the structures where CSF flows in the embryo:

1) As mentioned in our original response to the reviewers: the observation of diffusion of molecules in the CSF by other paths connecting the telencephalic and diencephalic ventricles to the central canal in zebrafish embryos had been observed before:

A characterization of the conduction path between the diencephalic ventricle and the central canal was reported in Fame et al., 2016 as well as Kondrychin et al., 2013, and to a lesser extent: Ribeiro et al., Open Biol. 2017 and Guo et al., 2018 where the same structures are visible but not commented on. These four key publications have been uploaded to the *eLife* website as related material.

The novelty of our study lies in i) the measurement of CSF flow in these conduction paths (Figure 7) and ii) the relevance for growth of the embryo during development (Figure 8).

2) The experiments requested by reviewer 2 "conventional histology, immunofluorescence staining and the analysis of marker genes"have already been done for the spinal cord in zebrafish embryos:

a) Kondrychyn, I., Teh, C., Sin, M., and Korzh, V. (2013). Stretching Morphogenesis of the Roof Plate and Formation of the Central Canal. PLoS ONE, 8(2), 1-12.

b) Guo et al. (Apical Cell-Cell Adhesions Reconcile Symmetry and Asymmetry in Zebrafish Neurulation. 2008. Iscience, 3, 63-85)

If we were not explicit enough on the former evidence for these structures in our revision, these articles are now heavily-cited when we refer to the conduction paths of the CSF in zebrafish embryo.

3) We agree with reviewer 2 on the fact that the walls of the ventricles may be in contact with the floor plate for what we had described as 'tel- and di-encephalospinal canals' ventrally and roof plate for what we had referred to as 'rhombencephalospinal canal' dorsally in the developing brain and spinal cord, with a transient occlusion due to lateral walls getting in close contact, which could explain the absence of CSF flow. That's exactly our view as well.

Since we understand the concern of the reviewer regarding the choice of the word “canal”, we now edited the manuscript to refer to as “conduction path” bridging between the ventricles and the central canal referring to the 4 publications mentioned above for the histological description of these structures.

If the last version of our manuscript was not clear enough, we apologize and now provide a documented description of the previous findings with a possible interpretation for these conduction paths in the Discussion of the revised manuscript.

We added references to the studies performed in mouse and chicken suggested by the reviewer in the Discussion.

[Editors' note: further revisions were suggested prior to acceptance, as described below.]

[…]The manuscript has been improved but there are some remaining issues that need to be addressed before acceptance, as outlined below. The critical issues have been adequately addressed and the reviewers believe the work will receive considerable attention in the field. They however identified a few remaining points where simple text revisions will be required before acceptance, as outlined below. Overall, the authors are asked to eliminate the term canal and explain that further work is needed to clarify its geometry.

Thank you for your feedback. As requested, we implemented these changes in the revised manuscript: we removed the terms canals and conduction paths when describing the complex circulatory system where CSF flows, and stated that future work will be necessary to clarify the geometry of the cavities ventral to the ventricles in the brain and dorsal than the central canal in the spinal cord.

Reviewer #2:Nature of the report:As stated by the authors: we disagree on this point. Although the authors use state-of-the-art technology to visualize and model cilia and CSF flow in the developing zebrafish ventricular system, the main character of the study is descriptive; it lacks a conceptual advance, i.e. it remains unclear what the physiological meaning of bi-directionality is and how the specific pattern of flow relates to any ciliopathies.

Our work is indeed descriptive: we use modelling from experts in fluid mechanics and quantitative flow measurements in vivoto establish how bidirectional flow is established in the central canal. We show how the bidirectional flow enables fast transport across the CSF of particles above 20 nm, and demonstrate that disruption of the connection from the brain ventricles to the central canal leads to reduction in growth during embryogenesis.

Anatomy in question:Unfortunately, the revised manuscript still lacks clarity. It now seems obvious that the structures the authors report – "conduction paths" (termed "canals" in the former versions of the manuscript, a term that persists in the revised manuscript as well) – represent the dorsal and ventral aspects of the developing ventricular system. This should be clearly stated in the manuscript, not only given as a possible explanation in the Discussion. Canals and conduction paths are confusing and imprecise; in addition, such terms tend to stick, which would be inappropriate. The cited paper by the Sive lab shows a parasagittal section, which does not resolve whether or not these structures are part of the ventricle and provides no proof for the existence of novel conduction paths. In addition, the Ribeiro et al. paper nicely shows that during transition of the neural tube primitive lumen into the central canal, the lumen becomes very narrow, but remains continuous between its dorsal and ventral aspects. If the authors insisted that their conduction paths and canals were independent entities, 3D imaging or morphing of histological sections and segmentation of the ventricular system would be required.

As requested by the reviewing editor, we removed the terms canals and conductions paths to describe the dorsal and ventral openings of the developing ventricular system in Figure 7: see text below.

“Embryonic CSF circulates between the brain ventricles and the central canal via a complex circulatory system

To understand how the CSF carries particles between CC and the brain ventricles at early stages, we imaged all the CSF-filled cavities by injecting the small dye Texas Red-Dextran (3,000 MW) in live 30 hpf embryos (Figure 7). […]During development, as it has already been partly described in rodents and zebrafish (Secv et al., 2009, Kondrychyn et al., 2013, Ribeiro et al., 2017), dorsal and ventral openings of the CC pL converge ventrally at 3 dpf (Figure 7F) to finally merge into a single central canal lumen at 5 dpf (Figure 7G).”